# Neurally-constrained modeling of human gaze strategies in a change blindness task

**Akshay Jagatap**[1¤a], **Simran Purokayastha**[1¤b], **Hritik Jain**[1¤c],
**Devarajan Sridharan**[1,2]*

**1** Centre for Neuroscience, Indian Institute of Science, Bangalore, India, **2** Computer Science and Automation, Indian Institute of Science, Bangalore, India

¤a Current address: IN Machine Learning, Amazon, World Trade Centre, Bangalore, India
¤b Current address: Department of Psychology, New York University, New York, United States of America
¤c Current address: The Data Science Institute, Columbia University, New York, United States of America
* sridhar@iisc.ac.in

**Data Availability Statement:** Data availability. Data associated with all figures and tables presented in the manuscript is available online at: https://doi.org/10.6084/m9.figshare.8247860. Code availability. Code for reproducing the all figures and

## Abstract

Despite possessing the capacity for selective attention, we often fail to notice the obvious. We investigated participants' (n = 39) failures to detect salient changes in a change blindness experiment. Surprisingly, change detection success varied by over two-fold across participants. These variations could not be readily explained by differences in scan paths or fixated visual features. Yet, two simple gaze metrics–mean duration of fixations and the variance of saccade amplitudes–systematically predicted change detection success. We explored the mechanistic underpinnings of these results with a neurally-constrained model based on the Bayesian framework of sequential probability ratio testing, with a posterior odds-ratio rule for shifting gaze. The model's gaze strategies and success rates closely mimicked human data. Moreover, the model outperformed a state-of-the-art deep neural network (DeepGaze II) with predicting human gaze patterns in this change blindness task. Our mechanistic model reveals putative rational observer search strategies for change detection during change blindness, with critical real-world implications.

## Author summary

Our brain has the remarkable capacity to pay attention, selectively, to important objects in the world around us. Yet, sometimes, we fail spectacularly to notice even the most salient events. We tested this phenomenon in the laboratory with a change-blindness experiment, by having participants freely scan and detect changes across discontinuous image pairs. Participants varied widely in their ability to detect these changes. Surprisingly, two low-level gaze metrics—fixation durations and saccade amplitudes—strongly predicted success in this task. We present a novel, computational model of eye movements, incorporating neural constraints on stimulus encoding, that links these gaze metrics with change detection success. Our model is relevant for a mechanistic understanding of human gaze strategies in dynamic visual environments.

tables presented in the manuscript is available online at: https://doi.org/10.6084/m9.figshare.8247860.

**Funding:** All the awards are received by Dr. Devarajan Sridharan (DS). The sponsors/funders and the corresponding grant numbers are listed below: Wellcome Trust-Department of Biotechnology India Alliance Intermediate fellowship – IA/I/15/2/502089 Science and Engineering Research Board Early Career award – ECR/2016/000403 Pratiksha Trust award – FG/SMCH-19-2047 India-Trento Programme for Advanced Research (ITPAR) grant – INT/ITAL Y/ITPAR-IV/COG/2018/G Department of Biotechnology-Indian Institute of Science Partnership Program grant Sonata Software foundation grant Tata Trusts grant The funders had no role in study design, data collection and analysis, decision to publish, or preparation of the manuscript.

**Competing interests:** I have read the journal's policy and the authors of this manuscript have the following competing interests: Devarajan Sridharan is a research consultant at Google. All other authors have declared that no competing interests exist.

## Introduction

We live in a rapidly changing world. For adaptive survival, our brains must possess the ability to identify relevant, changing aspects of our environment and distinguish them from irrelevant, static ones. For example, when driving down a busy road it is critical to identify changing aspects of the visual scene, such as vehicles shifting lanes or pedestrians crossing the street. Our ability to identify such critical changes is facilitated by visual attention–an essential cognitive capacity that selects the most relevant information in the environment, at each moment in time, to guide behavior [1].

Yet, our capacity for attention possesses key limitations. One such limitation is revealed by the phenomenon of "change blindness", in which observers fail to detect obvious changes in a sequence of visual images with intervening discontinuities [2,3]. Previous literature suggests that observers' lapses with detecting changes occur if the changes fail to draw attention; for example if the change is presented concurrently with distracting events, such as an intervening blank or transient noise patches. Change blindness, therefore, provides a useful framework for studying visual attention mechanisms and its lapses [4]. Such lapses have important real-world implications: observers' success in change blindness tasks has been linked to their driving experience levels [5,6] and safe driving skills [7].

In the laboratory, change blindness is tested, typically, by presenting an alternating sequence of (a pair of) images that differ in one important detail (Fig 1A, "flicker" paradigm) [2,3]. Participants are instructed to scan the images, with overt eye movements, to locate and identify the changing object or feature. While many previous studies have investigated the phenomenon of change blindness itself [8–10], very few studies have directly identified gaze-related factors that determine observers' success in change blindness tasks [4]. In this study, we tested 39 participants in a change blindness experiment with 20 image pairs (Fig 1A). Surprisingly, participants differed widely (by over 2-fold) in their success with detecting changes.

To understand the reason for these striking differences in performance, first, we analyzed participants' eye movement data, acquired at high spatial- and temporal- resolution, as they scanned each pair of images. We discovered that two key gaze metrics–mean fixation duration and the variance in the amplitude of saccades–were consistently predictive of participants' success. Next, we developed a model of overt visual search based on the Bayesian framework of sequential probability ratio testing [11–14] (SPRT), in which subjects decided the next, most probable location for making a saccade based on a posterior odds ratio test. In our SPRT model, we also incorporated biological constraints on stimulus encoding and transformation, based on well-known properties of the visual processing pathway [15,16] (e.g. bounded firing rates, Poisson variance, foveal magnification, and saliency computation).

Our neurally-constrained model mimicked key aspects of human gaze strategies in the change blindness task: model success rates were strongly correlated with human success rates, across the cohort of images tested. In addition, the model exhibited systematic variation in change detection success with fixation duration and saccade amplitude, in a manner closely resembling human data. Finally, the model outperformed a state-of-the-art deep neural network (DeepGaze II [17]) in predicting probabilistic patterns in human saccades in this change blindness task. We propose our model as a benchmark for mechanistic simulations of visual search, and for modeling human observer strategies during change detection tasks.

## Results

### Fixation and saccade metrics predict change detection success

39 participants performed a change blindness task (Fig 1A). Each experimental session consisted of a sequence of trials with a different pair of images tested on each trial. Images

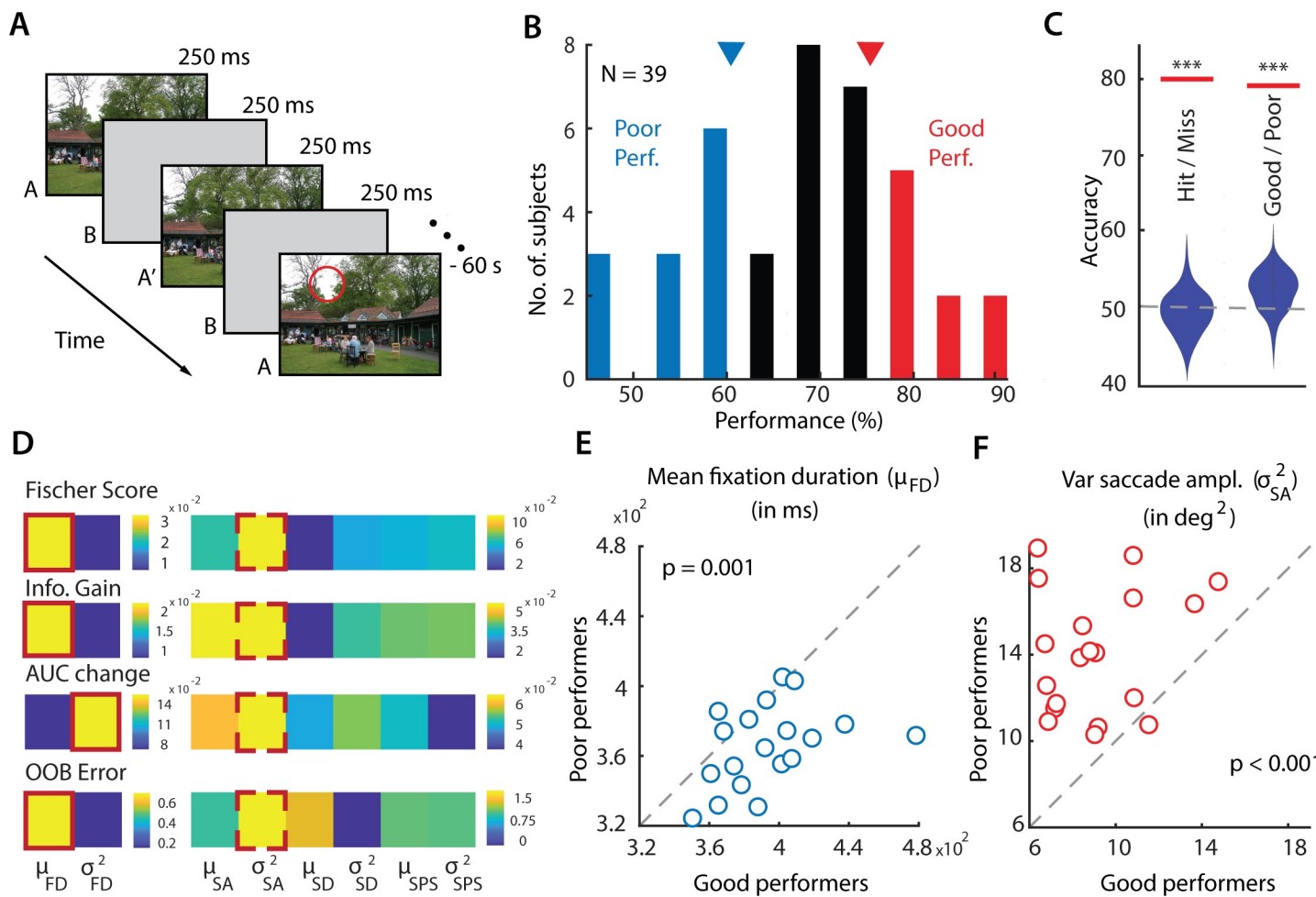

**Fig 1. Gaze metrics predict success in a change blindness experiment. A.** Schematic of a change blindness experiment trial, comprising a sequence of alternating images (A, A'), displayed for 250 ms each, with intervening blank frames (B) also displayed for 250 ms ("flicker" paradigm), repeated for 60 s. Red circle: Location of change (not actually shown in the experiment). All 20 change image pairs tested are available in Data Availability link. **B.** Distribution of success rates of n = 39 participants in the change blindness experiment. Red and blue bars: good performers (top 30th percentile; n = 9) and poor performers (bottom 30th percentile; n = 12), respectively. Inverted triangles: Cut-off values of success rates for classifying good (red) versus poor (blue) performers. **C.** Classification accuracy, quantified with area-under-the-curve (AUC), for classifying trials as hits versus misses (left horizontal line) and performers as good versus poor (right horizontal line), obtained with a support vector machine classifier. Violin plots: Null distributions of classification accuracies based on a permutation test (*** p<0.001). Error bars: Clopper-Pearson binomial confidence intervals. **D.** Feature selection measures for identifying the most informative features that distinguish good from poor performers. From top to bottom: Fisher score, Information gain, Change in area-under-the-curve (AUC) and bag of decision trees (for details, see Feature Selection Metrics in the Materials and Methods). Brighter colors indicate more informative features. Solid red outline: most informative feature in the fixation feature subgroup (left); dashed red outline: most informative feature in the saccade feature subgroup (right). FD—fixation duration, SA—saccade amplitude, SD—saccade duration, SPS—saccade peak speed. μ and $\sigma^2$ denote mean and variance of the respective parameter. **E.** Distribution of mean fixation duration ($\mu_{FD}$, in milliseconds) across 19 change images for good performers (x-axis) versus poor performers (y-axis); one change image pair, successfully detected by all performers, was not included in these analyses (see text). Each data point denotes average value of $\mu_{FD}$, across each category of performers, for each image tested. Dashed diagonal line: line of equality. p-value corresponds to significant difference in mean fixation duration between good and poor performers. **F.** Same as in E, but comparing variance of saccade amplitudes (in squared degrees of visual angle) for good versus poor performers. Other conventions are the same as in panel E.

presented included cluttered, indoor or outdoor scenes (see Data Availability link). To ensure uniformity of gaze origin across participants, each trial began when subjects fixated continuously on a central cross for 3 seconds. This was followed by the presentation of the change blindness image pair: alternating frames of two images, separated by intervening blank frames (250 ms each, Fig 1A). Of the image pairs tested, 20 were "change" image pairs, in that these differed from each other in one of three key respects (S1 Table): (i) size of an object changing; (ii) color of an object changing or (iii) change involving the appearance (or disappearance) of

an object. The remaining (either 6 or 7 pairs; Materials and Methods) were "catch" image pairs, which comprised an identical pair of images; data from these "catch" trials were not analyzed for this study (Materials and Methods; complete change image set in Data Availability link). Change- and catch- image pairs were interleaved and tested in the same pseudorandom order across subjects. Subjects were permitted to freely scan the images to detect the change, for up to a maximum of 60 seconds per image pair. They indicated having detected the change by foveating at the location of change for at least 3 seconds. A response was marked as a "hit" if the subject was able to successfully detect the change within 60 seconds, and was marked as a "miss" otherwise.

We observed that participants varied widely in their success with detecting changes: success rates varied over two-fold–from 45% to 90%–across participants (Fig 1B). These differences may arise from innate differences in individual capacities for change detection as well as other experimental factors (see Discussion). Nonetheless, we tested if individual-specific gaze strategies when scanning the images could explain these variations in change detection success.

First, we ranked subjects in order of their change detection success rates. Subjects in the top 30th (n = 9) and bottom 30th (n = 12) percentiles were labelled as "good" and "poor" performers, respectively (Fig 1B). This choice of labeling ensured robust differences in performance between the two classes: change detection success for good performers varied between 75% and 90%, whereas that for and poor performers varied between 45% and 61%. Nevertheless, the results reported subsequently were robust to these cut-offs for selecting good and poor performers (see S1 Fig for results based on performance median split). Next, we selected four gaze metrics from the eye-tracker: saccade amplitude, fixation duration, saccade duration and saccade peak speed (justification in the Materials and Methods) and computed the mean and the variance of these four metrics for each subject and trial. These eight quantities were employed as features in a classifier based on support vector machines (SVM) to distinguish good from poor performers (Materials and Methods). One image pair (#20), for which all participants correctly detected the change, was excluded for these analyses (Figs 1–3).

Classification accuracy (area-under-the-curve/AUC) for distinguishing good from poor performers was 79.9% and significantly above chance (Fig 1C, p<0.001, permutation test, Materials and Methods). We repeated these same analyses, but this time classifying each trial as a hit or miss. Classification accuracy was 77.7% and, again, significantly above chance (Fig 1C, p<0.001). Taken together, these results indicate that fixation- and saccade- related gaze metrics contained sufficient information to accurately classify change detection success.

Next, we identified gaze metrics that were the most informative for classifying good versus poor performers. This analysis was done separately for the fixation and saccade metric subsets: these were strongly correlated within each subset and uncorrelated across subsets (S2A Fig). For each metric, we performed feature selection with four approaches–Fisher score [18], AUC change [19] and Information Gain [20] and bag of decision trees (OOB error) [21]. A higher value of each selection measure reflects a greater importance of the corresponding gaze metric for classifying between good and poor performers. Among fixation metrics, mean fixation duration was assigned higher importance based on three out of the four feature selection measures (Fig 1D, solid red outline). Among the saccade metrics, variance of saccade amplitudes was assigned highest importance, based on all four feature selection measures (Fig 1D, dashed red outline). We confirmed these results *post hoc*: mean fixation duration was significantly higher for good performers, across images (Fig 1E; p = 0.0015, Wilcoxon signed rank test), whereas variance of saccade amplitude was significantly higher for poor performers (Fig 1F; p<0.001, Wilcoxon signed rank test).

We considered the possibility that the differences in fixation duration and saccade amplitude variance between good and poor performers could arise from differences in multiple,

distinct modes of these, respective distributions. Nonetheless, statistical tests provided no significant evidence for multimodality in either fixation duration or saccade amplitude distributions for either class of performers (S2B Fig) (Hartigan's dip test for unimodality; fixation duration: p>0.05, in 8/9 good performers with median p = 0.74, and in 8/12 poor performers with median p = 0.31; saccade amplitudes: p>0.05, in 9/9 good performers with median p = 0.99 and in 10/12 poor performers with median p = 0.99).

In sum, these results indicate that two key gaze metrics–mean fixation duration and variance of saccade amplitude–were strong and sufficient predictors of change detection success in a change blindness experiment.

Next, we tested if more complex features of eye movements–such as scan paths, fixation maps or fixated object features–differed systematically between good and poor performers.

Scan path data is challenging to compare across individuals because scan paths can vary in terms of both the number and sequence of image locations samples. We compared scan paths across participants by encoding them into a "string" sequences (Materials and Methods). Briefly, fixation points for each image were clustered, with data pooled across subjects, and individual subjects' scan paths were encoded as strings based on the sequence of clusters visited across successive fixations (Fig 2A and 2B). We then quantified the deviation between scan paths for each pair of subjects using the edit distance [22]. Median scan path edit distances were not significantly different between good and poor performer pairs (Fig 2C, p = 0.14, Wilcoxon signed rank test). We also tested if the median inter-category edit distance between the good and poor performer categories would be higher than the median intra-category edit distance among the individual (good or poor) performer categories (Fig 2D). These edit distances were also not significantly different (p>0.1, one-tailed signed rank test).

Second, we asked if fixation "maps"–two-dimensional density maps of the distribution of fixations [23]–were different across good and poor performers. For each image, we correlated fixation maps across every pair of participants (Materials and Methods). Again, we observed no significant differences between fixation map correlations between good- and poor- performer pairs (Fig 2E, p = 0.29, Wilcoxon signed rank test), nor significant differences between intra-category (good vs. good and poor vs. poor) fixation map correlations and inter-category (good vs. poor) correlations (Fig 2F, p>0.1, one-tailed signed rank test).

Third, we asked if overall statistics of saccades were different across good and poor performers. For this, we computed the probabilities of saccades between specific fixation clusters ("domains"), ordered by the most to least fixated locations on each image (Materials and Methods). The saccade probability matrix, estimated by pooling scan paths across each category of participants, is shown in Fig 3A (average across n = 19 image pairs). Visual inspection of the saccade probability matrices revealed no apparent differences between the good and poor performers (difference in S3A Fig). In addition, we tested if we could classify between good and poor performers based on individual subjects' saccade probability matrices. Classification accuracy with an SVM based on saccade probability matrix features (~56.67%, Fig 3B) was not significantly different from chance (p>0.1, permutation test).

Fourth, we tested whether good and poor performers differed in terms of fixated image features, as estimated with principal components analysis (Fig 3C, Materials and Methods). These fixated features typically comprised horizontal or vertical edges at various spatial frequencies, and were virtually identical between good and poor performers (Fig 3D, first six principal features for each class). We observed significant correlations across components of identical rank between good and poor performers (median r = 0.22, p<0.001, across top n = 150 components that explained ~80% of the variance). Similar correlations were obtained with fixated features obtained with the saliency map [24] (median r = 0.20, p<0.001, S3B Fig).

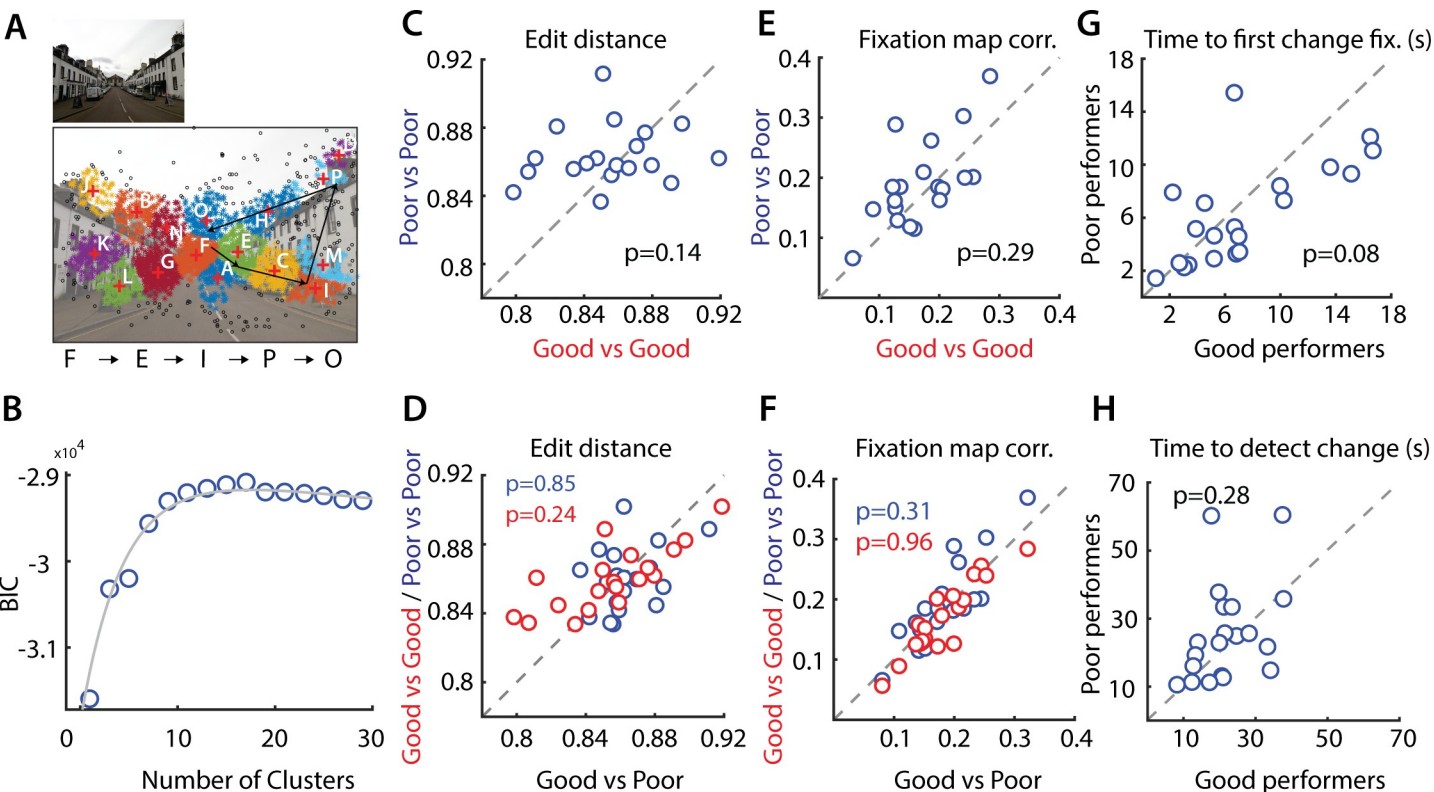

**Fig 2. Scan paths and fixation maps do not distinguish good from poor performers. A.** (Left) Representative image used in the change blindness experiment (Image #6 in Data Availability link). (Right) Clustering of the fixation points based on the peak of the fitted BIC (n = 13) profile. Fixation points in different clusters are plotted in different colors. Black fixations occurred in fixation sparse regions that were not included in the clustering. Black arrows show a representative scan path–a sequence of fixation points. The character "string" representation of this scan path is denoted on the right side of the image. **B.** Variation in the Bayesian Information Criterion (BIC; y-axis) with clustering fixation points into different numbers of clusters (x-axis; Materials and Methods). Circles: Data points. Gray curve: Bi-exponential fit. **C.** Distribution of edit distances among good performers (x-axis) versus edit distances among poor performers (y-axis). Each data point denotes median edit distance for each image tested (n = 19). Other conventions are the same as in Fig 1E. **D.** Distribution of intra-category edit distance (y-axis), among the good or among the poor performers, versus the inter-category edit distance (x-axis), across good and poor performers. Red and blue data: intra-category edit distance for good and poor performers respectively. Each data point denotes the median for each image tested (n = 19). Other conventions are the same as in panel C. **E.** Same as panel C, but comparing Pearson correlations of fixation maps among good (x-axis) and poor performers (y-axis). Other conventions are the same as in panel C. **F.** Same as panel D, but comparing intra- versus inter-category Pearson correlations of fixation maps. Other conventions are the same as in panel D. **G.** Distribution of time to first fixation within the region of change (in seconds) for good performers (x-axis) versus poor performers (y-axis). Other conventions are the same as in panel C. **H.** Same as in E, but comparing time to detect change (in seconds) for good versus poor performers. Other conventions are the same as in panel G.

Fifth, we tested whether good and poor performers differed systematically in the spatial distributions of fixations relative to the change location, before change was detected. For this, we computed the frequency of fixations and the total fixation duration, based on the distance of fixation relative to the center of the change location (binned in concentric circular windows of increasing radii, in steps of 50 pixels, Materials and Methods). We observed no systematic differences in the distributions of either total fixation duration, or frequency of fixations, relative to the change location between good and poor performers (S4 Fig; p = 0.99 for fixation duration, p = 0.97 for fixation frequency, Kolmogorov-Smirnov test). In other words, the spatial distribution of fixations, relative to the change location, was similar between good and poor performers.

Finally, we tested whether good and poor performers differed in the time to first fixation on the region of change, or the time to detect changes (on successful trials). Again, we observed no significant differences in the distributions of either time to first fixation, or time to detect changes, between good and poor performers (Fig 2G and 2H; p = 0.08 for time to first fixation

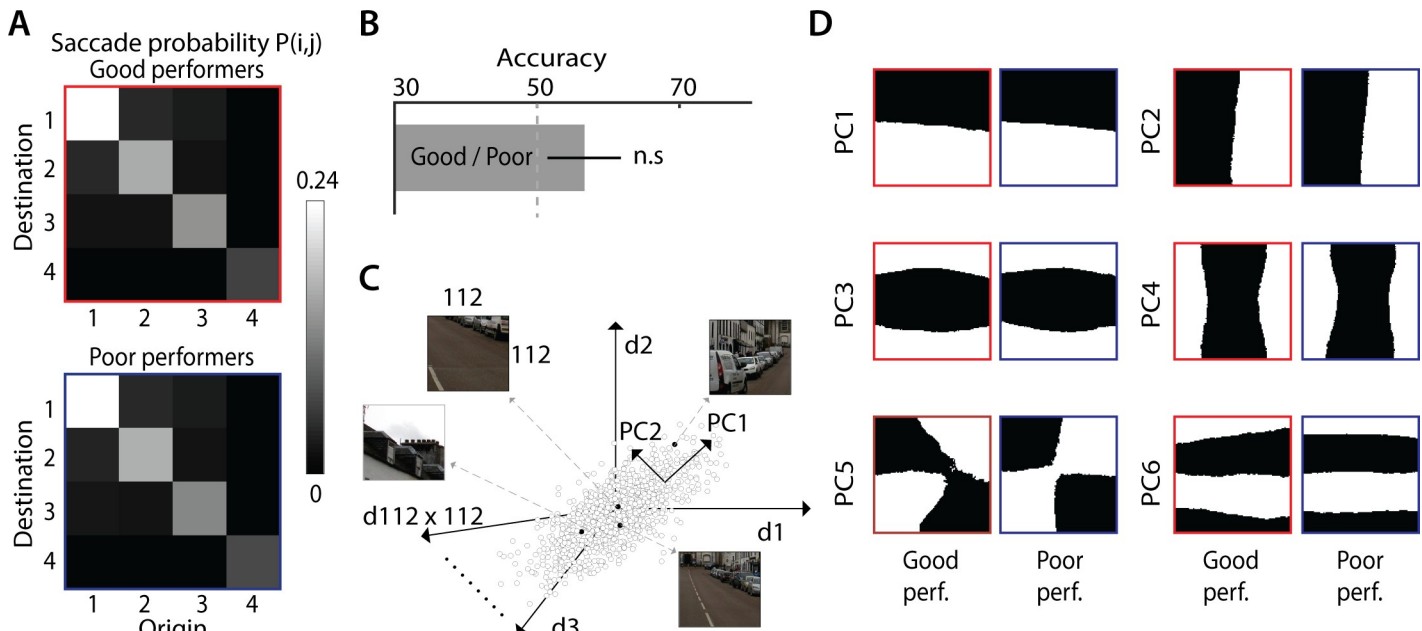

**Fig 3. Saccade probabilities and fixated features are similar across good and poor performers. A.** Average saccade probability matrices for the good performers (top; red outline) and poor performers (bottom; blue outline). These correspond to probabilities of making a saccade between different "domains" (1–4), each corresponding to a (non-contiguous) collection of image regions, ordered by frequency of fixations: most fixated regions (domain 1) to least fixated regions (domain 4). Cell (I, j) (row, column) of each matrix indicates the probability of saccades from domain j to domain i. **B.** Classification accuracy for classifying good versus poor performers based on the saccade probability matrix features, using a support vector machine classifier. Other conventions are as in Fig 1D. Error bars: s.e.m. **C.** Identifying low-level fixated features across good and poor performers. 112x112 image patches were extracted, centered around each fixation, for each participant; each point in the 112x112 dimensional space represents one such image patch. Principal component analysis (PCA) was performed to identify low-level spatial features explaining maximum variance among the fixated image patches, separately for good and poor performers. **D.** Top 6 principal components, ranked by proportion of variance explained, corresponding to spatial features explaining greatest variance explained across fixations, for good performers (left panels) and poor performers (right panels). These spatial features were highly correlated across good and poor performers (median r = 0.20, p<0.001, across n = 150 components).

in change region, p = 0.28 for time to detect change, signrank test). Taken together with the previous analysis, these results indicate that poor performers fixated as often and as close to areas near the change, but simply failed to detect these changes successfully.

Overall, these analyses indicate that relatively simple gaze metrics like fixation durations and saccade amplitudes predicted successful change detection. More complex metrics like scan paths, fixated image features or the spatial distribution of fixations, were not useful indicators of change detection success. In other words, "low-level" gaze metrics, rather than "high-level" scanning strategies, determined participants' success with change detection.

## A neurally-constrained model of eye movements for change detection

We developed a neurally-constrained model of change detection to explain these empirical trends in the data. Briefly, our model employs the Bayesian framework of Sequential Probability Ratio Testing (SPRT) framework [14,15] to simulate rational observer strategies when performing the change blindness task. We incorporated key neural constraints, based on known properties of stimulus encoding in the visual processing pathway, into the model. For ease of understanding we summarize key steps in our model's saccade generation pipeline (Fig 4A and 4B), first; a detailed description is provided thereafter.

In the model, distinct neural populations, with (noisy) Poisson firing statistics, encode the saliency of the foveally-magnified image at each region. During fixation, following each alternation (Fig 1A, either A followed by A', or vice versa) the model computes a posterior odds

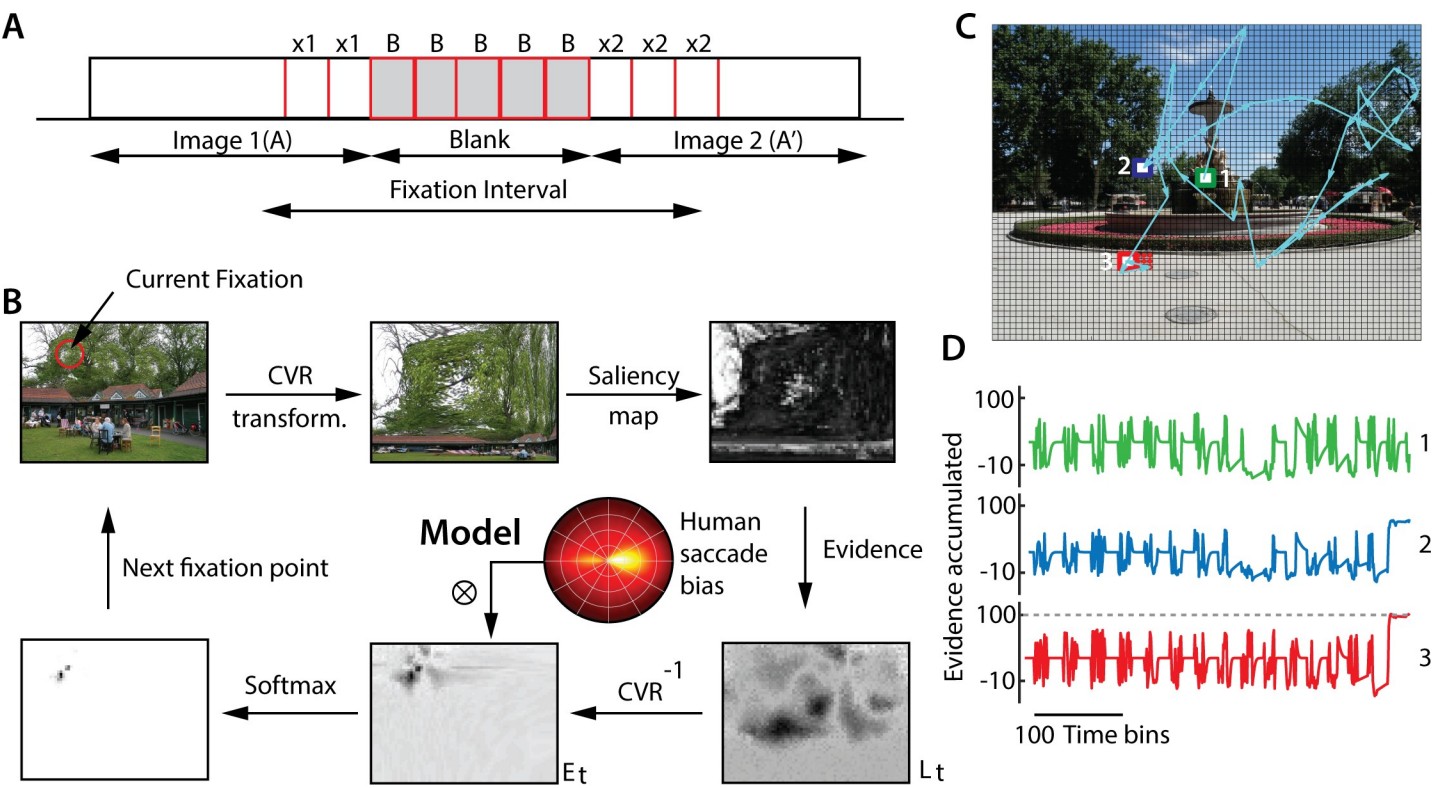

**Fig 4. A Bayesian model of gaze strategies for change detection. A.** Schematic showing a typical fixation across the pair of images (A, A') and an intervening blank. **B.** Detailed steps for modeling change detection (see text for details). (Clockwise from top left) At each fixation, a Cartesian variable resolution (CVR) transform is applied to mimic foveal magnification, followed by a saliency map computation to determine firing rates at each location. Instantaneous evidence for change versus no change (log-likelihood ratio, log L(t)) is computed across all regions of the image. An inverse CVR transform is applied to project the evidence back into the original image space, where noisy evidence is accumulated, (sequential probability ratio test, drift-diffusion model). The next fixation point is chosen using a softmax function applied over the accumulated evidence ($E_t$). To model human saccadic biases, a distribution of saccade amplitudes and turn angles is imposed on the evidence values prior to selecting the next fixation location (polar plot inset). **C.** A representative gaze scan path following model simulation (cyan arrows). Colored squares: specific points of fixation (see panel D). Grid: Fine divisions over which the image was sub-divided to facilitate evidence computation. Green (1), blue (2) and red (3) squares denote first (beginning of simulation), intermediate (during simulation) and last (change detection) fixation points, respectively. **D.** Evidence accumulated as a function of time for the same three representative regions as in panel D; each color and number denotes evidence at the corresponding square in panel C. When the model fixated on the green or blue squares (in panel C), the accumulated evidence did not cross the threshold for change detection. As a result, the model continued to scan the image. When the model fixated on the red square (in the change region), the accumulated evidence crossed threshold (horizontal, dashed gray line) and the change was detected.

ratio for change versus no change at each region and at each instant of time (Eq 1), and accumulates this ratio as "evidence" (Eq 2, Results). If the accumulated evidence exceeds a predetermined (positive) threshold for change detection at the location of fixation, the model is deemed to have detected the change. If, the accumulated evidence dips below a predetermined (negative) threshold for "no change" at the fixated location, the observer terminates the current fixation. The next fixation location is chosen based on a stochastic (softmax) decision rule (Eq 3), with the probability of saccade to a region being proportional to the accumulated evidence at that region. Note that both images—odd and even—must be included in these computations to generate each saccade. The model continues scanning over the images in the sequence until either the change is detected or until the trial duration has elapsed (as in our experiment), whichever occurs earlier.

**Neural representation of the image pair.** At the onset of each fixation, the image was magnified foveally based on the center of fixation [25], with the Cartesian Variable Resolution (CVR) transform [26] (Materials and Methods; S5 Fig). Next, a saliency map was computed with the frequency-tuned salient region detection method [24] for each image of each pair.

**Table 1. Model parameters and their default values.**

| Parameters | Symbol | Value | Description |
|---|---|---|---|
| Time bin | $\Delta t$ | 25 ms | Unit of time for the model |
| Image duration | $\tau$ | $10\Delta t$ | Duration for which each image or blank is shown |
| Trial duration | – | 60 s | Total duration of trial |
| Temperature | T | 0.01 | Modulates stochasticity of next saccade |
| Decay factor | $\gamma$ | 0.004 | Decay of the evidence with time (inversely related) |
| Decay scale | $\beta$ | 4.0 grid units | Spatial range of evidence decay |
| Noise scale | W | U(-5, 5) | Models noise in evidence accumulation |
| Prior odds ratio | P | 0.1 | Prior odds of change to no change |
| Change threshold | $F_c$ | 100 | Threshold to determine change |
| "No change" threshold | $F_n$ | -20 | Threshold to determine "no change". |
| Threshold decay | $\zeta$ | 0 | Decay rate of no-change threshold |
| Foveal magnification factor | FMF | 0.05 | Magnification of the fixated region on the fovea according to the CVR transform |
| Firing rate bounds | $\lambda_{min}, \lambda_{max}$ | 5, 120 spikes/bin | Minimum and maximum firing rates |
| Firing rate prior | $\mu_f$ | 3 spikes/bin | Expected difference in firing rates |

Saliency computation was performed on the foveally-magnified image, rather than on the original image, to mimic the sequence of these two computations in the visual pathway; we denote these foveally-transformed saliency values as S and S', for each image (A) and its altered version (A'), respectively.

Each image was partitioned into a uniform 72x54 grid of equally-sized regions. We index each region in each image pair as $A_1, A_2 \ldots, A_N$ and $A'_1, A'_2 \ldots, A'_N$, respectively (N = 3888). Distinct, non-overlapping, neural populations encoded the saliency value ($S_i, S_i'$) in each region of each image. While in the brain, neural receptive fields typically overlap, we did not model this overlap here, for reasons of computational efficiency (Materials and Methods). The firing rates for each neural population were generated from independent Poisson processes. The average firing rate for each region $\lambda_i$ was modeled as a linear function of the average saliency of image patch falling within that region as: $\lambda_i(S_i) = \lambda_{min} + (\lambda_{max} - \lambda_{min})\langle S_i^k \rangle_k$, where $S_i^k$ is the saliency value of the kth pixel in region $A_i$, and the angle brackets denote an average across all pixels in that region. In other words, when the change between images A and A' occurred in region i, the difference in firing rates between $\lambda_i$ and $\lambda'_i$ was proportional to the difference in saliency values across the change.

We modeled each change detection trial (total duration T, Table 1), as comprising of a large number of time bins of equal duration ($\Delta t$, Table 1). At every time bin, the number of spikes from each neural population was drawn a Poisson distribution whose mean was determined by the average saliency of all pixels within the region. At the end of each fixation, the model either indicated its detection of change, thereby terminating the simulation, or shifted gaze to a new location. The precise criteria for signaling change versus shifting gaze are described next.

For ease of description, we depict a typical fixation in Fig 4A. The first image of the pair (say, A) persists m time-bins from the onset of the current fixation. Next, a blank epoch occurs from m+1 to p time-bins. Following this, the second image of the pair (A') appears for an interval from p+1 to n time bins, until the end of fixation. We denote the number of spikes produced by neural population i at time t by $\chi_t^i$. $X^i$ and $Y^i$ represent the total number of spikes produced by neural population i when fixating at the first and second images respectively, during the current fixation. Thus, $X^i = \sum_{t=1}^{m}(\chi_t^i)$; $Y^i = \sum_{t=p+1}^{n}(\chi_t^i)$. We denote the number of spikes in the blank period as $B^i = \sum_{t=m+1}^{p}\chi_t^i = 0$. For simplicity, we assume that no spikes

occurred during the blank period ($B^i = 0$), although this is not a strict requirement, as the key model computations rely on relative rather than absolute firing rates. In sum, the observer must perform change detection with a noisy neural representation derived from saliency map of the foveally-magnified image.

**Modeling change detection with an SPRT rule.** The observer faces two key challenges with change detection in this change blindness task. First, were the images not interrupted by a blank, a simple pixel-wise difference of firing rates over successive time epochs would suffice to localize the change. For example, computing $|\langle X^i \rangle - \langle Y^i \rangle|$ (where |x| denotes the absolute value of x, and angle brackets denote average over many time bins), and testing if this difference is greater than zero at any region $i$, suffices to identify the location of change. On the other hand, such an operation does not suffice when images are interleaved with a blank, as in change blindness tasks. For example, a pixel-wise subtraction of each image from the blank ($|\langle X^i \rangle - \langle B^i \rangle|$ or $|\langle Y^i \rangle - \langle B^i \rangle|$) yields large values at all locations of the image. Therefore, when images are interrupted by a blank, information about the first image must be maintained across the blank interval and compared with second image following the blank, for detecting the change. Second, even if no blank occurred between the images, a pixel-wise differencing operation would not suffice, due to the stochasticity of the neural representation: a non-zero difference in the number of spikes from a particular region, $i$ ($|X^i - Y^i|$) is not direct evidence of change at that location. In other words, the observer's strategy for this change blindness task must take into account both the occurrence of the blank between the two images, as well as the, stochasticity in the Poisson neural representation of the image, for successfully detecting changes.

To address both of these challenges, we adopt an SPRT-based search rule. First, we compute the difference in the number of spikes between the first and second image at each region $A_i$ in the image. We denote the random variable indicating this difference by $Z^i = X^i - Y^i$, and its value at end of time bin $t$ as $z$. We then compute a likelihood ratio for change (C) versus no change (N), as:

$$L_i(t;\ z) = \frac{p(Z^i(t) = z | C)}{p(Z^i(t) = z | N)} \qquad (1)$$

Specifically, the observer tests if the observed value of $Z^i$ was more likely to arise from two generating processes (Change, C), or could from a single, underlying generating process (No Change, N). This computation is performed at each time step following the onset of the second image ($t > p$) of each pair. Details of computing this likelihood ratio for Poisson processes are provided in the Materials and Methods; for our model this computation involves an infinite sum, which we calculate using Bessel functions and efficient analytic approximations [27]. The functional form of the log-likelihood ratio resembles a piecewise linear function of firing rate differences (S6A and S6B Fig, see next section), which can be readily achieved by the output of simple neural circuits [28–30].

Second, the observer integrates the "evidence" for change at location $A_i$, by accumulating the logarithm of the likelihood ratio $\log(L_i(t))$, along with the log of the prior odds ratio ($P_i$), as in the SPRT framework.

$$E_i(t) = (1 - \gamma_i(t))E_i(t - 1) + \log(L_i(t)) + \log(P_i) + W_i(t) \qquad (2)$$

where $\gamma_i \in [0,1]$ is a decay parameter for evidence accumulation at location $A_i$, which simulates "leaky" evidence accumulation [15,31] with larger values of $\gamma_i$, indicating greater "leak" in evidence accumulation, $P_i$ is the prior odds ratio of change to no change (P(C)/P(N)) at each location, $W_i(t)$ represents white noise, sampled from a uniform distribution (Table 1), to mimic

noisy evidence accumulation [32]. Here we assume that the prior ratio is constant across time and space, but nonetheless study the effect of varying prior ratios on model performance (next section). Both of these features–leak and noise in evidence accumulation–are routinely incorporated in models of human decision-making [31], and are grounded in experimental observations in brain regions implicated in decision-making [15]. Evidence accumulation occurs in the original, physical space of the image, and not in the CVR transformed space (Fig 4B). Note that this formulation of an SPRT decision involves evaluating and integrating fully the logarithm of the Bayesian posterior odds ratio (product of the prior odds and likelihood ratio, $P_i \times L_i(t)$).

Evidence accumulation is performed for each region in the image; $E_i(t)$ for each region is calculated independently of the other regions. If the accumulated evidence $E_i(t)$ crosses a positive threshold, $F_c$ (Table 1), the observer stops scanning the image and region $A_i$, at which the threshold $F_c$ was crossed, is declared the "change region". If, on the other hand, the accumulated evidence crosses a negative (no-change) threshold $F_n$ (Table 1), the observer terminates the current fixation and determines the next region to fixate, $A_k$, based on a softmax probability function:

$$p_k = e^{\frac{E_k}{T}} / \Sigma_{i=1}^{N} e^{\frac{E_i}{T}} \tag{3}$$

where $E_i$ is the evidence value for region $i$, $N$ is the number of regions in the image, and T is a temperature parameter which controls the stochasticity of the saccade (decision) policy (Materials and Methods; see also next section). For selecting the next point of gaze fixation, we also matched directional saccadic biases typically observed in human data [33] (Fig 4B, described in Materials and Methods section on "Comparison of model performance with human data"). In some simulations we also decayed the no-change threshold ($F_n$) with different decay rates ($\zeta$; Table 1) and studied its effect on model performance. Because we observed virtually no false alarms (signaling a no-change location as change) in our experimental data (0.06% of all trials; Materials and Methods) we did not model decay in the change threshold ($F_c$), which would have yielded significantly more false alarms.

Note that although we have not explicitly modeled inhibition-of-return (IOR), this feature emerges naturally from the evidence accumulation rule in the model. Following each fixation, the accumulated evidence for no-change decays gradually (Eq 2), thereby reducing the probability that subsequent fixations occur, immediately, at the erstwhile fixated location. This feature encourages the model to explore the image more thoroughly. We illustrate gaze shifts by the model in an exemplar change blindness trial (Fig 4C and 4D). The model's scan path is indicated by cyan arrows showing a sequence of fixations, ultimately terminating at the change region. When the model fixated, initially, on regions with no change (Fig 4C, squares with green/1 or blue/2 outline), transient evidence accumulation occurred either favoring a change (positive fluctuations) or favoring no change (negative fluctuations) (Fig 4D, green and blue traces, respectively). In each case, evidence decayed to baseline values rapidly during the blank epochs, when no new evidence was available, and the accumulated evidence did not cross threshold. Finally, when the model fixated on the change region (Fig 4C, square with red outline/3), evidence for a change continued to accumulate, until a threshold-crossing occurred (Fig 4D, red trace, threshold: dashed gray line). At this point, the change was deemed to have been detected, and the simulation was terminated.

## Model trends resemble qualitative trends in human experimental data

We tested the effect of key model parameters on change detection performance, to test for qualitative matches with our experimental findings. We simulated the model and measured

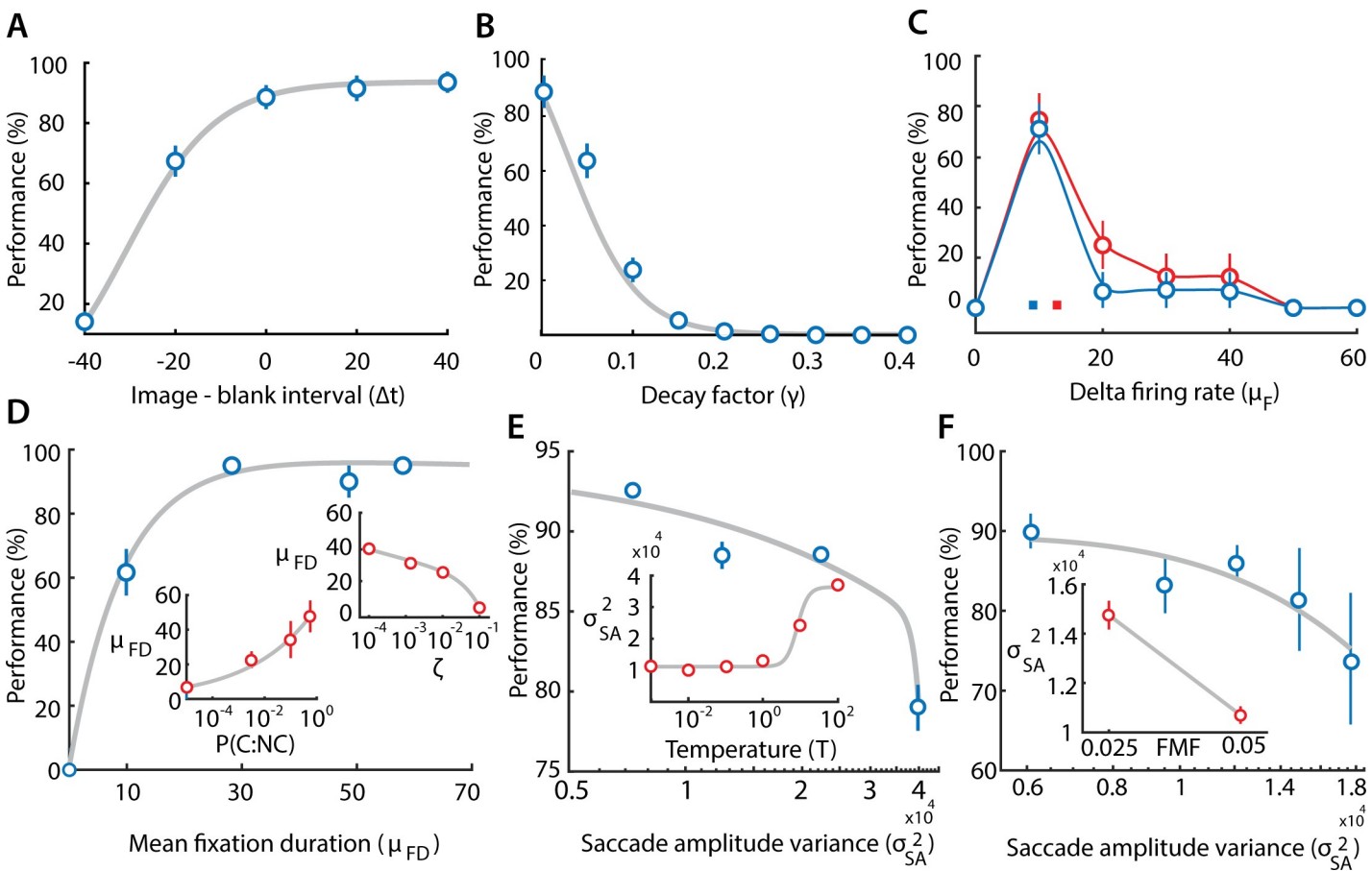

**Fig 5. Effect of model parameters on change detection success. A.** Change in model performance (success rates, % correct) with varying the relative interval of the images and blanks, measured in units of time bins (Δt = 25 ms/time bin; Table 1), while keeping the total image+blank interval constant (at 50 time bins). Positive values on the x-axis denote larger image intervals, as compared to blanks, and vice versa, for negative values. Blue points: Data; gray curve: sigmoid fit. **B.** Same as in panel (A), but with varying the maximum decay factor (γ; Eq 2). Curves: Sigmoid fits. **C.** Same as in panel (A) but with varying the firing rate prior (μf) for image pairs with the lowest (blue; bottom 33rd percentile) and highest (red; top 33rd percentile) magnitudes of firing rate changes. Curves: Smoothing spline fits. Colored squares: μf corresponding to the center of area of the two curves. **D.** Same as in panel (A), but with varying the mean fixation duration ($\mu_{FD}$; measured in time bins, Δt = 25 ms/time bin). (Inset; lower) Variation of $\mu_{FD}$ with prior ratio of change to no change (P(C:NC)). (Inset; upper) Same as lower inset but with varying threshold decay rate ζ (Table 1). **E.** Same as in panel (A), but with varying saccade amplitude variance ($\sigma^2_{SA}$). (Inset) Variation of $\sigma^2_{SA}$ with the softmax function temperature parameter (T) (see text for details). **F.** Same as in panel (A), but with varying saccade amplitude variance ($\sigma^2_{SA}$). (Inset) Variation of $\sigma^2_{SA}$ with the foveal magnification factor (FMF). Other conventions in B-F are the same as in panel A. Error bars (all panels): s.e.m.

change detection performance by varying each model parameter in turn (Table 1, default values), while keeping all other parameters fixed at their default values. For these simulations we employed the frequency-tuned salient region detection method [24] to generate the saliency map. The first three simulations (Fig 5A–5C) tested whether the model performed as expected based on its inherent constraints. The last three simulations (Fig 5D–5F) evaluated whether emergent trends in the model matched empirical observations regarding gaze metrics in our study (Fig 1E and 1F). The results reported represent averages over 5–10 repetitions of each simulation.

First, we tested the effect of varying the relative durations of the image and the blank, while keeping their overall presentation duration (image+blank) constant. Note that no new evidence accrues during the blanks, whereas decay of accumulated evidence continues. Therefore, extending the duration of the blanks, relative to the image, should cause a substantial deterioration in the performance of the model. The simulations confirmed this hypothesis:

performance deteriorated (or improved) systematically with decreasing (or increasing) durations of the image relative to the blank (Fig 5A).

Second, we tested the effect of varying the magnitude of the decay factor ($\gamma$, Table 1). Decreasing $\gamma$ prolongs the (iconic) memory for evidence relevant to change detection; $\gamma = 1$ represents no memory (immediate decay; no integration) of past evidence, whereas $\gamma = 0$ indicates reliable memory (zero decay; perfect integration) of past evidence (refer Eq 2). Model success rates were at around 80% for $\gamma = 0$ and performance degraded systematically with increasing $\gamma$ (Fig 5B); in fact, the model was completely unable to detect change for $\gamma$ values greater than around 0.2, suggesting the importance of the transient memory of the image across the blank for successful change detection.

Third, we tested the effect of varying $\mu_f$, the prior on the magnitude of the difference between the firing rates (across the image pair) in the change region (Fig 5C). For this, we divided images into two extreme subsets (highest and lowest 1/3rd), based on a tercile (three-way) split of firing rate magnitude differences. The performance curve for the highest tercile (largest firing rate differences in change region) of images was displaced rightward relative to the performance curve for the lowest tercile (smallest firing rate differences). Specifically, $\mu_f$ corresponding to the center of area of the performance curves was systematically higher for the images with higher firing rate differences (Fig 5C, colored squares).

Fourth, we tested the effect of varying mean fixation duration ($\mu_{FD}$)–a key parameter identified in this study as being predictive of success with change detection. The mean fixation duration is not a parameter of the model. We, therefore, varied the mean fixation duration, indirectly, by varying the prior odds ratio ($P$) and the decay rate ($\zeta$) of the no-change threshold ($F_n$). A lower prior odds ratio of change to no-change biases evidence accumulation toward the no-change threshold, leading to shorter fixations (and vice versa; Fig 5D, lower inset). On the other hand, a higher decay rate of the no-change threshold leads to a greater probability of bound crossing of the evidence in the negative direction, again leading to shorter fixations (and vice versa; Fig 5D; upper inset). In either case, we found that decreasing (increasing) the mean fixation duration produced systematic deterioration (improvement) in the performance of the model (Fig 5D). These results recapitulate trends in the human data, indicating that increased fixation duration may be a key gaze metric indicating change detection success.

Fifth, we tested the effect of varying the saccade amplitude variance ($\sigma_{SD}^2$)–the other key parameter we had identified as being predictive of change detection success. Again, because the variance of the saccade amplitude is not a parameter of the model, we varied this, indirectly, by varying the temperature ($T$) parameter in the softmax function: a higher temperature value leads to random sampling from many regions of the image, thereby increasing $\sigma_{SD}^2$ whereas a low temperature value leads to more deterministic sampling, thereby reducing $\sigma_{SD}^2$ (Fig 5E, inset). With increasing saccade variance, performance dropped steeply (Fig 5E).

Finally, we also explored the effect of varying the foveal magnification factor (FMF) across a two-fold range. Saccade amplitude variance decreased robustly as the FMF increases (Fig 5F, inset) (see Discussion). As with the previous simulation, we observed a systematic decrease in performance with increasing saccade amplitude variance (Fig 5F), again, recapitulating trends in the human data.

Taken together, these results show that gaze metrics that were indicative of change detection success in the change blindness experiment also systematically influenced change detection performance in the model. Specifically, the two key metrics indicative of change detection success in humans, namely, fixation duration and variance of saccade amplitude, were also predictive of change detection success in the model. These effects could be explained by changing specific, latent parameters in the model (e.g. decay rate of the no-change threshold, prior

ratios, foveal magnification factors). Our model, therefore, provides putative mechanistic links between specific gaze metrics and change detection success in the change blindness task.

## Model performance mimics human performance quantitatively

In addition to these qualitative trends, we sought to quantify similarities between model and human performance in this change blindness task. For this analysis, we modeled biases inherent in human saccade data (S7 Fig) by matching key saccade metrics in the model–amplitude and turn angle of saccades–with human data (Figs 6A and 7A, r = 0.822, p<0.01; see Materials and Methods section on "Comparison of model performance with human data"). For these simulations, and subsequent comparisons with a state-of-the-art deep neural network model (DeepGaze II) [17] we used the saliency map generated by the DeepGaze network rather than

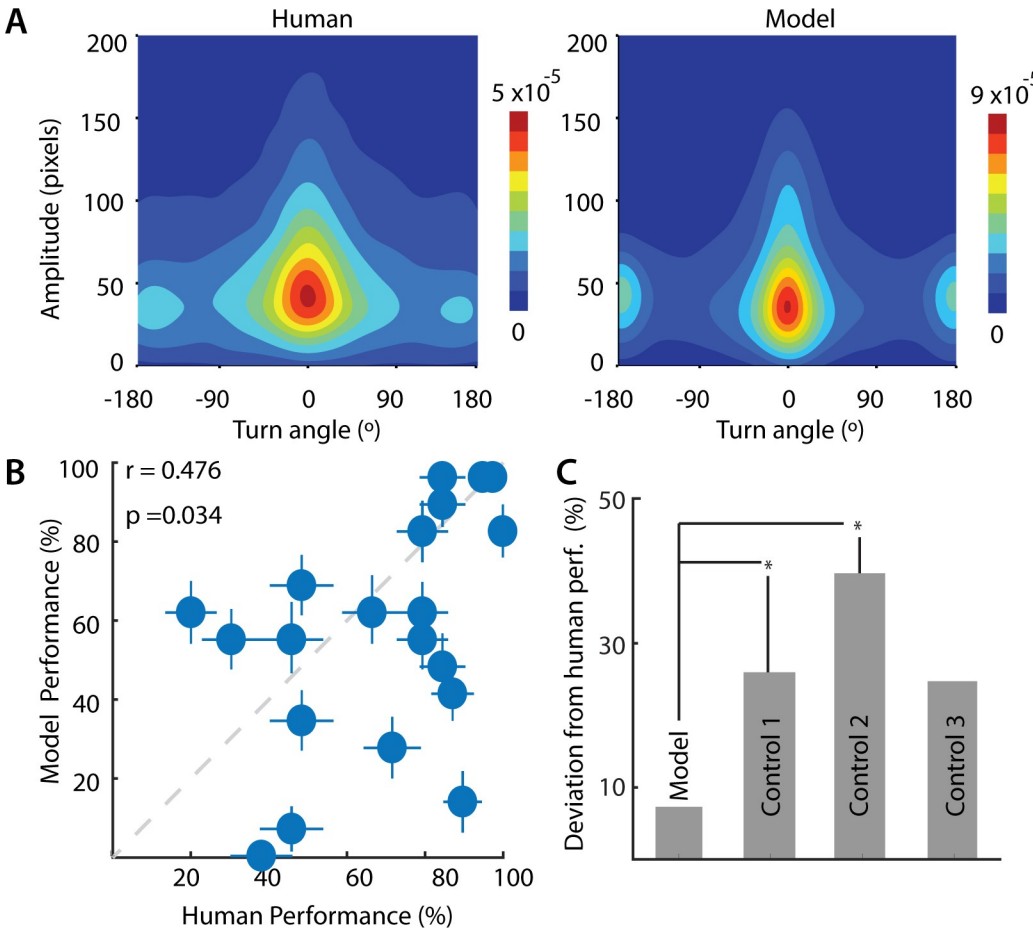

**Fig 6. Comparison between human and model performance. A.** (Left) Joint distribution of saccade amplitude and saccade turn angle for human participants (averaged over n = 39 participants). Colorbar: Hotter colors denote higher proportions. (Right) Same as in the left panel, but for model, averaged over n = 40 simulations. **B.** Correlation between change detection success rates for human participants (x-axis) and the model (y-axis). Each point denotes average success rates for each of the 20 images tested, across n = 39 participants (human) or n = 40 iterations (model). Error bars denote standard error of the mean across participants (x-axis) or simulations (y-axis). Dashed gray line: line of equality. **C.** Average absolute deviation from human performance of the sequential probability ratio test (SPRT) model (Model, leftmost bar), for a control model in which evidence decayed rapidly (Control 1, γ = 1; second bar from left), for a control model in which the stopping rule was based on the derivative of the posterior odds ratio (Control 2; third bar from left), or for a control model which employed a random search strategy (Control 3, T = 10⁴; rightmost bar). p-values denote significance levels following a paired signed rank test, across n = 20 images (*p < 0.05).

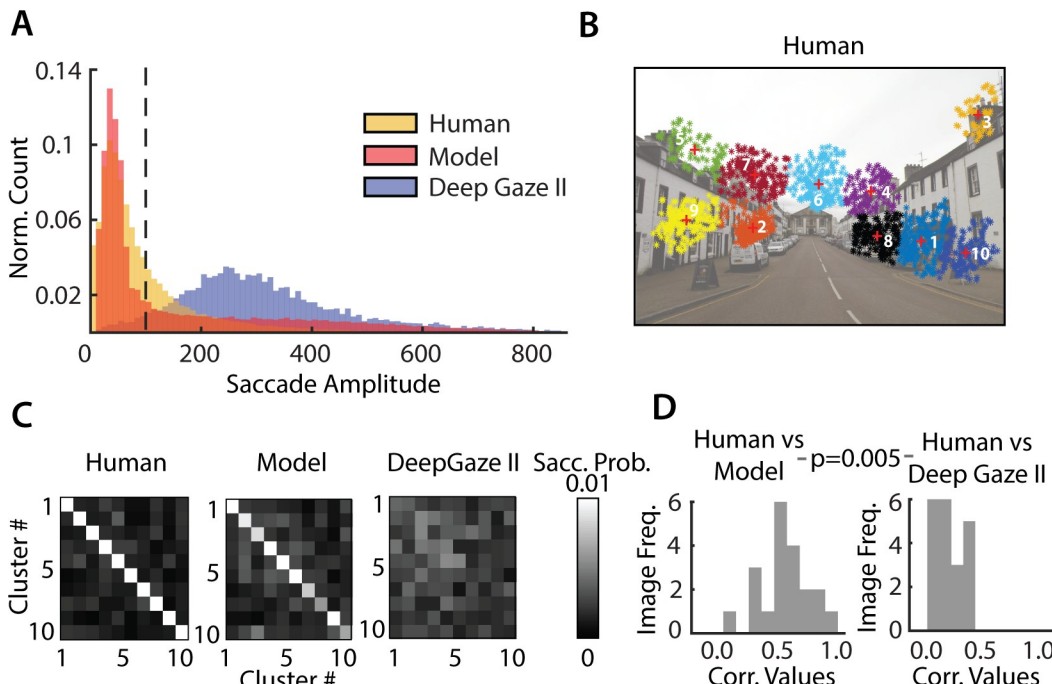

**Fig 7. Comparison between human, model and Deep Gaze II performance. A.** Distribution of saccade amplitudes for human participants (yellow), sequential probability ratio test (SPRT) model (red) and the Deep Gaze II neural network (blue). **B.** Top 10 clusters of human fixations, ranked by cumulative fixation duration (rows/columns 1–10). Increasing indices correspond to progressively lower cumulative fixation duration. **C.** Saccade probability matrix (left) averaged across all images and all participants, (middle) for simulations of the sequential probability ratio test (SPRT) model, and (right) for the Deep Gaze II neural network. **D.** Distribution, across images, of the correlations (r-values) of saccade probability matrices between human participants and sequential probability ratio test (SPRT) model (left) and human participants and Deep Gaze II neural network (right). p-value indicates pairwise differences in these correlations across n = 20 images.

the frequency-tuned salient region detection algorithm, so as to enable a direct comparison between our model and DeepGaze.

As a first quantitative comparison, we tested whether image pairs in which human observers found difficult to detect changes (S2C Fig), were also challenging for the model. For this, we compared the model's success rates across images with observers' success rates in the change blindness experiment. Remarkably, the model's success rates, averaged across 40 independent runs, correlated significantly with human observers' average success rates (Fig 6B, r = 0.476, p = 0.034, robust correlations across n = 20 images).

We compared the Bayesian SPRT search rule, as specified in our model, against three alternative control models, each with a different search strategy or stopping rule: (i) a model in which evidence decayed rapidly, so that the decision to signal change was based on the instantaneous posterior odds ratio alone; (ii) a model in which the stopping rule was based on crossing a threshold "rate of change" of the posterior odds ratio, and (iii) a model that employed a random search strategy (Materials and Methods). For each of these models, the average absolute difference in performance with the human data was significantly higher, compared with that of the original model (Fig 6C; p<0.05 for 2/3 control models; Wilcoxon signed-rank test). Moreover, none of the control model's success rates correlated significantly with human observers' success rates (r = 0.09–0.42, p>0.05, for all 3/3 control models; robust correlations).

Finally, we tested whether model gaze patterns would match human gaze patterns beyond that achieved by state-of-the-art fixation prediction with a deep neural network: DeepGaze II

[17]. First, we quantified human gaze patterns by computing the probability of saccades pairwise among the top 10 clusters with the largest number of fixations (e.g. Fig 7B) for each image. We then compared these human saccade probability matrices (Fig 7C, left) with those derived from simulating the model (Fig 7C, middle) as well as with those generated by the DeepGaze network (Fig 7C, right). For the latter, saccades were simulated using the same softmax rule as employed in our model (Eq 3) along with inhibition-of-return [34] (Materials and Methods); in addition, for each image, we identically matched the distribution of fixation durations between DeepGaze and our model (Materials and Methods).

The model's saccade probability matrix (Fig 7C, middle) closely resembled the human saccade probability matrix (Fig 7C, left), indicating that model was able to mimic human saccades patterns closely. On the other hand, the DeepGaze saccade matrix (Fig 7C, right) deviated significantly from the human saccade probability matrix. Confirming these trends, we observed significantly higher correlations between the human saccade probabilities and our model's saccade probabilities (Fig 7D, left) as compared to those with DeepGaze's saccade probabilities (Fig 7D, right) (human-SPRT model: median r = 0.51, human-DeepGaze II: median r = 0.14; p<0.01 for significant difference in correlation values across n = 20 images, signed rank test). These results were robust to the underlying saliency map in our SPRT model: replacing DeepGaze's saliency map with the frequency-tuned salient region detection method yielded nearly identical results (S8A and S8B Fig).

The chief reason for these differences was readily apparent upon examining the saccade amplitude distributions across the human data, our SPRT model and DeepGaze: whereas the human and model distributions contained many short saccades, the DeepGaze distribution contained primarily long saccades (Fig 7A). Consequently, we repeated the comparison of saccade probabilities limiting ourselves to the range of saccade amplitudes in the DeepGaze model. Again, we found that our model's saccade probabilities were better correlated with human saccade probabilities (S8C and S8D Fig) (human-SPRT model: median r = 0.29, human-DeepGaze II: median r = 0.10; p<0.001). We propose that these differences occurred because DeepGaze saccades are generated based on relative saliencies of different regions across the image, whereas saliency computation, *per se*, may be insufficient to model human saccade strategies in change blindness tasks or, in general, in change detection tasks.

In summary, change detection success rates were robustly correlated between human participants and the model. Moreover, our model outperformed a state-of-the-art deep neural network in predicting gaze shifts among the most probable locations of human gaze fixations in this change blindness task.

## Discussion

The phenomenon of change blindness reveals a remarkable property of the brain: despite the apparent richness of visual perception, the visual system encodes our world sparsely. Stimuli at locations to which attention is not explicitly directed are not effectively processed [4]. Even salient changes in the visual world sometimes fail to capture our attention and remain undetected. Visual attention, therefore, plays a critical role in deciding the nature and content of information that is encoded by the visual system.

In a laboratory change blindness experiment, we observed that participants varied widely in their ability to detect changes. These differences cannot be directly attributed to participants' inherent change detection abilities. Nevertheless, a recent study evaluated test-retest reliability in change blindness tasks, and found that observers' change detection performance was relative stable over periods of 1–4 weeks [35]. In our study, participants whose fixations lasted marginally longer, on average, and whose saccades were less spatially variable, were best able

to detect changes. Given the intricate link between mechanisms for directing eye-movements and those governing visual attention [36–38], our results suggest the hypothesis that spatial attention shifts more slowly in time (higher fixation durations), and less erratically in space (lower saccade variance), in order to enable participants to detect changes effectively.

To explain our experimental observations mechanistically, we developed, from first principles, a neurally-constrained model based on the Bayesian framework of sequential probability ratio testing [15,31]. Such SPRT evidence accumulation models have been widely employed in modelling human decisions [31], and also appear to have a neurobiological basis [15]. In our model, we incorporated various neural constraints including foveal magnification, saliency maps, Poisson statistics in neural firing and human saccade biases. Even with these constraints, the model was able to faithfully reproduce key trends in the human change detection data, both qualitatively and quantitatively (Figs 5 and 6). The model's success rates correlated with human success rates, and the model reproduced key saccade patterns in human data, outperforming competing control models (Figs 6B and 6C, 7C and S8).

On the one hand, our study follows a rich literature on human gaze models, that fall, loosely, into two classes. The first class of "static" models use information in visual saliency maps [23,37,39] to predict gaze fixations. These saliency models, however, do not capture dynamic parameters of human eye fixations, which are important for understanding strategies underlying visual exploration in search tasks, like change blindness tasks. The second class of "dynamic" models seek to predict the temporal sequence of gaze shifts [40–43]. Nevertheless, these approaches were developed for free-viewing paradigms, and comparatively few studies have focused on gaze sequence prediction during search tasks [44,45]. On the other hand, several previous studies have developed algorithms to address the broader problem of "change point" detection [46–48]. Yet, none of these algorithms are neurally-constrained (e.g. foveal magnification, Poisson statistics), and none models gaze information or saccades. To the best of our knowledge, ours is the first neurally-constrained model for gaze strategies in change blindness tasks, and developing and validating such a model is a central goal of this study.

Specifically, our model outperformed a state-of-the-art deep neural network (DeepGaze II), in terms of predicting saccade patterns in this change blindness task. Yet, a key difference must be noted when comparing our model with DeepGaze. Our model relies on a decision rule based on posterior odds for generating saccades: For this, it must compare evidence for change versus no change across the two images. In our simulations, in contrast, the DeepGaze model generates saccades independently on the two images, without comparing them. Based on these simulations, we found that our model's gaze patterns provided a closer match to human data compared to gaze patterns from DeepGaze (Figs 7C and 7D and S8). Because DeepGaze is a model tailored for predicting free-viewing saccades, this comparison serves only to show that even a state-of-the-art free-viewing saliency prediction algorithm is not sufficient to accurately predict gaze patterns in the context of a change detection (or change blindness) task. In other words, saccades made with the goal of detecting changes are likely to be different from saccades made in free-viewing conditions.

Our model exhibited several emergent behaviors that matched previous reports of human failures in change blindness tasks. First, the model's success rates improved systematically as the blank interval was reduced (Fig 5A); this trend mimics previously-reported patterns in human change blindness tasks, in which shortening the interval of the intervening blank improves change detection performance [4]. Second, the model's success rates improved systematically with reducing the evidence decay rate across the blank (Fig 5B). In other words, retaining information across the blank was crucial to change detection success. This result may have intriguing links with neuroscience literature, which has shown that facilitating neural activity in oculomotor brain regions (e.g. the superior colliculus) during the blank epoch

counters change blindness [49]. Third, the model's ability to detect changes improved when its internal prior (expected firing rate difference) aligns with the actual firing rate difference at the change region (Fig 5C). These results may explain a results from a previous study [50], which found that familiarity with the context of the visual stimulus was predictive of change detection success.

Finally, the model provided mechanistic insights about key trends observed in our own experiments, specifically, a critical dependence of success rates on mean fixation durations and the variance of saccade amplitudes (Fig 5D, 5E and 5F). Fixation durations in the model varied systematically either with altering the prior odds ratio or the decay rate of the no-change bound. Note that the prior odds ratio corresponds to an individual's prior belief in the prior probability of change to no change. The lower this ratio, the higher the degree of belief in no change, and the sooner the individual seeks to break each fixation. In our model, this was achieved by having the prior ratio bias evidence accumulation toward the no-change (negative) bound. Similarly, faster decay of the no-change bound, possibly reflecting a stronger "urgency" to break fixations, resulted in faster bound crossing and, therefore, shorter fixations. Regardless of the mechanism, shorter fixation durations resulted in impaired change detection performance (Fig 5E), providing a putative mechanistic link between fixation durations and change detection success in the experimental data. In addition, saccade amplitude variance modulated systematically with changes in the foveal magnification factor (FMF). With higher foveal magnification the model is, perhaps, able to better distinguish features in regions proximal to the fixation location, and saccade to them, thereby resulting in overall shorter saccades, and lower variance. Moreover, the higher foveal magnification, enables analyzing the region of change with higher resolution, thereby leading to better change detection performance. As a consequence performance degraded systematically with increased saccade amplitude variance (Fig 5F), the common underlying cause for each being the change in the foveal magnification factor. This provides a plausible mechanism for higher variance of saccade amplitudes in "poor" performers.

We implemented three control models in this study. The first control model—in which evidence decayed rapidly ($\gamma = 1$)—mimics the scenario of rapidly decaying short-term memory; this model signals the change based on threshold crossing of the instantaneous, rather than the accumulated, posterior odds. In the second control model, we employed an alternative stopping rule: a rapid, large change in the posterior odds ratio sufficed to signal the change. Such a "temporally local" stopping rule obviates the need for evidence accumulation (short-term memory) and may be implemented by neural circuits that act as temporal change detectors (differentiators). The third control model mimicked a random saccade strategy, with a high temperature parameter ($T = 10^4$) of the softmax function. This model establishes baseline (chance) levels of success, if an observer were to ignore model evidence and saccade randomly to different locations on each image, and arrive at the change region "by chance". Each of these control models fell short of our SPRT model in terms of their match to human performance.

Nonetheless, our SPRT model can be improved in a few ways. First, saliency maps in our model were typically computed with low-level features (e.g. Fig 5; the frequency tuned salient region method). Incorporating more advanced saliency computations (e.g. semantic saliency) [51] into the saliency map could render the model more biologically realistic. Second, although our neurally-constrained model provides several biologically plausible mechanisms for explaining our experimental observations, it does not identify which of these mechanisms is actually at play in human subjects. To achieve this objective, model parameters may be fit with maximum likelihood estimation [52] or Bayesian methods for sparse data (e.g. hierarchical Bayesian modelling)[53]. Yet, in its current form, such fitting is rendered challenging because

the model is not identifiable: multiple parameters in the model (e.g. prior ratios or decay of the no-change threshold) produce similar effects on specific gaze parameters (e.g. fixation durations, Fig 5D). Future extensions to the model, for example, by measuring and modeling more gaze metrics for constraining the model, may help overcome this challenge. Such model-fitting will find key applications for identifying latent factors contributing to inter-individual differences in change detection performance.

Our simulations have interesting parallels with recent literature. With a battery of cognitive tasks Andermane et al. [35] identified two factors that were critical for predicting change detection success: "visual stability"–the ability to form stable and robust visual representation– and "visual ability"–indexing the ability to robustly maintain information in visual short term memory. Other studies have identified associated psychophysical factors, including attentional breadth [54] and visual memory [55] as being predictive of change detection success. We propose that (higher) fixation durations and (lower) variability of saccade amplitudes may both index a (higher) "visual stability" factor, indexing the ability to form more stable visual representations. In contrast, the temporal decay factor (Table 1, $\gamma$) and spatial decay scale (Table 1, $\beta$) may correspond to visual memory and attentional breadth, respectively; each could comprise key components of the "visual ability" factor, indexing robust maintenance of information in short-term memory. Our model provides a mechanistic test-bed to systematically explore the contribution of each of these factors and their constituent components to change detection success in change blindness experiments.

A mechanistic understanding of the behavioral and neural processes underlying change blindness will have important real-world implications: from safe driving [56] to reliably verifying eyewitness testimony [57]. Moreover, emerging evidence suggests that change blindness (or a lack thereof) may be a diagnostic marker of neurodevelopmental disorders, like autism [58–60]. Our model characterizes gaze-linked mechanisms of change blindness in healthy individuals and may enable identifying the mechanistic bases of change detection deficits in individuals with neurocognitive disorders.

## Materials and methods

### Ethics statement

Informed written consent was obtained from all participants. The study was approved by the Institutional Human Ethics Committee (IHEC) at the Indian Institute of Science (IISc), Bangalore.

### Experimental protocol

We collected data from n = 44 participants (20 females; age range 18–55 yrs) with normal or corrected-to-normal vision and no known impairments of color vision. Of these, data from 4 participants, who were unable to complete the task due to fatigue or physical discomfort, were excluded. Data from one additional participant was irretrievably lost due to logistical errors. Thus, we analyzed data from 39 participants (18 females).

Images were displayed on a 19-inch Dell monitor at 1024x768 resolution. Subjects were seated, with their chin and forehead resting on a chin rest, with eyes positioned roughly 60 cm from the screen. Each trial began when subjects continually fixated on a central cross for 3 seconds. This was followed by presentation of the change image pair sequence for 60s: each frame (image and blank) was 250 ms in duration. The trial persisted until the subjects indicated the change by fixating at the change region for at least 3 seconds continuously ("hit"), or if the maximum trial duration (60 s) elapsed and the subjects failed to detect the change ("miss"). An online algorithm tracked, in real-time, the location of the subjects' gaze and signaled the

completion of a trial based on whether they were able to fixate stably at the location of change. Each subject was tested on either 26 or 27 image pairs, of which 20 pairs differed in a key detail (available in Data Availability link); we call these the "change" image pairs. The remaining image pairs (7 pairs for 30 subjects and 6 pairs for 9 subjects) contained no changes ("catch" image pairs); data from these image pairs were not analyzed for this study (except for computing false alarm rates, see next). To avoid biases in performance, the ratio of "change" to "catch" trials was not indicated to subjects beforehand, but subjects were made aware of the possibility of catch trials in the experiment. We employed a custom set of images, rather than a standardized set (e.g.[4]), due to the possibility that some subjects might have been familiar with change images used in earlier studies.

Overall, the proportion of false-alarms–proportion of fixations with durations longer than 3s in catch trials–was negligible (~0.06%, 17/32248 fixations across 264 catch trials) in this experiment. To further confirm if the subjects indeed detected the change on hit trials, a post-session interview was conducted in which each subject was presented with one of each pair of change images in sequence and asked to indicate the location of perceived change. The post-session interview indicated that about 5.7% (31/542) of hit trials were not recorded as such; in these cases, the total trial duration was 60 s indicating that even though the subject fixated on the change region, the online algorithm failed to register the trial as a hit. In addition, 2.9% (7/238) of miss trials, in which the subjects were unable to detect the location of change in the post-session interview, ended before the full trial duration (60 s) had elapsed; in these cases, we expect that subjects triggered the termination of the trial by accidentally fixating for a prolonged duration near the change. We repeated the analyses excluding these 4.8% (38/780) trials and observed results closely similar to those reported in the text. Finally, eye-tracking data from 0.64% (5/780) trials were corrupted and, therefore, excluded from all analyses.

Subjects' gaze was tracked throughout each trial with an iViewX Hi-speed eye-tracker (SensoMotoric Instruments Inc.) with a sampling rate of 500 Hz. The eye-tracker was calibrated for each subject before the start of the experimental session. Various gaze parameters, including saccade amplitude, saccade locations, fixation locations, fixation durations, pupil size, saccade peak speed and saccade average speed, were recorded binocularly on each trial, and stored for offline analysis. Because human gaze is known to be highly coordinated across both eyes, only monocular gaze data was used for these analyses. Each session lasted for approximately 45 minutes, including time for instruction, eyetracker calibration and behavioral testing.

## SVM classification and feature selection based on gaze metrics

We asked if subjects' gaze strategies would be predictive of their success with detecting changes. To answer this question, as a first step, we tested if we could classify good versus poor performers (Fig 1C) based on their gaze metrics alone. As features for the classification analysis, we computed the mean and variance of the following four gaze metrics: saccade amplitude, fixation duration, saccade duration and saccade peak speed recorded by the eyetracker. We did not analyze two other gaze metrics acquired from the eyetracker: saccade average speed and pupil diameter for these analyses. Saccade average speed was highly correlated with saccade peak speed across fixations (r = 0.93, p<0.001), and was a redundant feature. In addition, while pupil size is a useful measure of arousal [61], it is often difficult to measure reliably, because slight, physical movements of the eye or head may cause apparent (spurious) changes in pupil size that can be confounded with real size changes. Before analysis, feature outliers were removed based on Matlab's *boxplot* function, which considers values as outliers if they are greater than $q_3 + w \times (q_3 - q_1)$ or less than $q_1 - w \times (q_3 - q_1)$, where $q_1$ and $q_3$ are the 25th

and 75th percentiles of the data, respectively, and setting $w = 1.5$ provides 99.3 percentile coverage for normally distributed data. To avoid biases in estimating gaze metrics for good versus poor performers this last fixation at the change location (a minimum of 3 seconds of data) were removed from the eyetracking data for all "hit" trials before further analyses.

Following outlier removal, these eight measures were employed as features in a classifier based on support vector machines (SVM) to classify good from poor performers (*fitcsvm* function in Matlab). The SVM employed a polynomial kernel, and other hyperparameters were set using hyperparameter optimization (*OptimizeHyperParameters* option in Matlab). Features from each image were included as independent data points in feature space. Classifier performance was assessed with 5-fold cross validation, and quantified with the area-under-the-curve (AUC [62]). For these analyses, we included gaze data from all but one image (Image #20, see Data Availability link), in which every subject detected the change correctly. Significance levels (p-values) of classification accuracies were assessed with permutation testing by randomly shuffling the labels of good and poor performers across subjects 100 times and estimating a null distribution of classification accuracies; significance values correspond to the proportion of classification accuracies in the null distribution that were greater than the actual classification accuracy values. A similar procedure was used for SVM classification of trials into hits and misses except that, in this case, class labels were based on whether the trial was a hit or a miss, and permutation testing was performed by shuffling hit or miss labels across trials. Because we employed summary statistics (e.g. mean, variance) of the gaze metrics in these feature selection analysis, we tested for unimodality of the logarithm of the respective gaze metric distributions with Hartigan's dip test for unimodality [63].

Next, we sought to identify gaze metrics that best distinguished good from poor performers. For this we employed four standard metrics—Fisher score [18], AUC change [19] and Information Gain [20] and bag of decision trees [21]–which quantify the relative importance of each feature for distinguishing the two groups of subjects (Fig 1D). A detailed description of these metrics is provided next.

## Feature selection metrics

(i) *Fisher score* computes the "quality" of features based on their extent of overlap across classes. In a two-class scenario, Fisher Score for the $j^{th}$ feature is defined as,

$$F(j) = \frac{(\bar{x}_j^{(+)} - \bar{x}_j)^2 + (\bar{x}_j^{(-)} - \bar{x}_j)^2}{\left(\frac{1}{n^{(+)}-1}\right)\sum_{i=0}^{n^{(+)}}(x_{i,j}^{(+)} - \bar{x}_j^{(+)})^2 + \left(\frac{1}{n^{(-)}-1}\right)\sum_{i=1}^{n^{(-)}}(x_{i,j}^{(-)} - \bar{x}_j^{(-)})^2} \tag{4}$$

where, $\bar{x}_j$ is the average value of the $j^{th}$ feature. Similarly $\bar{x}_j^{(+)}$ and $\bar{x}_j^{(-)}$ are the average of $j^{th}$ feature for the positive and negative category respectively. Here $\bar{x}_{i,j}^{(+)}$ and $\bar{x}_{i,j}^{(-)}$ denote the $j^{th}$ feature of $i^{th}$ sample-index for each category, with n(+) and n(-) being the number of positive and negative instances respectively. A more discriminative feature has a higher Fisher score.

(ii) *AUC change* describes the change in area-under-the-curve (AUC) with removing each feature in turn. The AUC (A) is the area under the ROC curve, plotted by varying the discrimination threshold and plotting the True Positive Rate (TPR) as a function of the False Positive Rate (FPR).

$$A = \int_{x=0}^{1} TPR(FPR^{-1}(x))dx \tag{5}$$

A more discriminative feature's absence produces a higher deterioration in classification accuracy.

(iii) *Information gain* is a classifier-independent measure of the change in entropy upon partitioning the data based on each feature. A more discriminative feature has a higher information gain. Given binary class labels Y for a feature X, the entropy of Y (E(Y)) is defined as,

$$E(Y) = -p_+ \log(p_+) - p_- \log(p_-) \tag{6}$$

where, $p_+$ is the fraction of positive class labels and $p_-$ is the fraction of negative class labels.

The Information Gain given Y for a feature X is given by,

$$IG(X, Y) = E(Y) - \min_i \frac{n_{X>div(i)} E(Y_{X>div(i)}) + n_{X<div(i)} E(Y_{X<div(i)})}{n_{X>div(i)} + n_{X<div(i)}} \tag{7}$$

$$div(i) = \frac{X_{sorted}(i) + X_{sorted}(i+1)}{2}$$

where, $n_{X>div(i)}$ and $n_{X<div(i)}$ is the number of entries of X greater than and less than *div(i)*, $Y_{X>div(i)}$ and $Y_{X<div(i)}$ are the entries of Y for which the corresponding entries of X are greater than and less than *div(i)* respectively and $X_{sorted}(i)$ indicates a feature vector with its values sorted in ascending order. A more discriminative feature has a higher Information Gain.

(iv) Out-of-bag error based on a bag of decision trees is an approach for feature selection using bootstrap aggregation on an ensemble of decision trees. Rather than using a single decision tree this approach avoids overfitting by growing an ensemble of trees on independent bootstrap distributions drawn from the data. The most important features are selected by out-of-bag estimates of feature importance in the bagged decision trees (OOB error). We used the *Treebagger* function, as implemented in Matlab, with saccade and fixation features as inputs to the model, which classified if the data belonged to a good or poor performer. The number of trees was set to 6, with all other hyperparameters set to their default values.

## Analysis of scan paths and fixated spatial features

We compared scan paths and low-level fixated (spatial) features across good and poor performers. To simplify comparing scan paths across participants, we adopted the following approach: we encoded each scan path into a finite length string. As a first step, fixation maps were generated to observe where the subjects fixated the most. Very few fixations occurred in object-sparse regions (e.g. sky), or had uniform color or texture, like the walls of a building (Fig 2A). In contrast, many more fixations around crowded regions with more intricate details. For each image, fixation points of all subjects were clustered, and each cluster was assigned a character label. The entire scan path, comprising a sequence of fixations, was then encoded as a string of cluster labels.

Before clustering fixation points, we sought to minimize the contributions of regions with very low fixation density. To quantify this we adopted the following approach: Let $x_i$ be a fixation point and let $D_{x_i}^r$ denote the average Euclidean distance of $x_i$ from the set of other fixation points which are at a radius $r$ from it. Let $(D_{x_i}^r)^{-1}$ denote the inverse of $D_{x_i}^r$. Now, we distributed all the fixation points uniformly on the image; let $U$ denote this set. We find the point $y_i$ in $U$ that was closest (in Euclidean distance) to $x_i$, and compute $(D_{y_i}^r)^{-1}$, as before. Then, the fixation density at the fixation point $x_i$ was defined as $\rho(x_i) = (D_{x_i}^r)^{-1} / (D_{y_i}^r)^{-1}$. Thus, all points with density less than 1 indicate regions which were sampled with less density than that corresponding to a uniform sampling strategy. These fixation points with very low fixation density

were grouped into a single cluster since these occurred in regions that were explored relatively rarely. For these analyses *r* was set to 40 pixels, although the results were robust to variations of this parameter. The remaining fixation points were clustered using k-means clustering algorithm.

The main challenges in working with the k-means algorithm are with: (i) deciding the number of clusters (k) and ii) deciding initial cluster centers. To overcome these, we utilized the Bayesian Information Criterion (BIC) employed in the context of x-means clustering [64]: this allowed us to determine the optimum k. For each k ranging from 1 to 50, a BIC score was computed. Following smoothing, k corresponding to the highest BIC score was selected as the optimum cluster count. Once the number of clusters was fixed, the initial cluster centers were fixed using an iterative approach: For each iteration, initial cluster centers were selected using the *k-means++* algorithm [65] and the values which gave the highest BIC score were selected as the initial cluster centers. Using the k and initial centers identified with these approaches, the fixation points were clustered for each image (Fig 2A, right). Once these clusters were identified for each image, we employed four approaches for the analysis of scan-paths and fixated spatial features.

First, we computed the edit distance between scan paths [22]. Briefly, the edit distance provides an intuitive measure of the dissimilarity between two strings. It corresponds to the minimal number of "edit" operations—insertions, deletions or substitutions—that are necessary to transform one string into the other. For each image, the edit distance between the scan paths of each pair of subjects was calculated and normalized (divided) by the longer scan path length of the pair; this was done to normalize for differences in scan path length across subjects. A distribution of normalized edit distances was calculated among the good performers, and among the poor performers, across images. Median edit distance of each category of performers was compared against the other, with a Wilcoxon signed rank test. However, note that the lack of a significant difference would only indicate that good performers and poor performers, each, followed similarly-consistent strategies. Therefore, to test whether these strategies were indeed significantly different between good and poor performers, we compared the median edit distance among the good (or poor) performers (intra-category edit distance) with the median edit distance across good and poor performers (inter-category edit distance), for all images, with a one-tailed signed rank test.

Second, we computed the probabilities of making a saccade among specific types of clusters, which we call "domains". Clusters obtained for each image were sorted in descending order of cumulative fixation duration. These were then grouped into four "domains", based on quartiles of fixation duration, and ordered such that the first domain had the highest cumulative fixation duration (most fixated domain) and the last domain had the least cumulative fixation duration (least fixated domain). We then computed the probability of making a saccade from each domain to the other. We denote these saccade probabilities as: $P(i_k, j_{k+1})$, which represents the probability of making a saccade from domain *i* at fixation *k* to domain *j* at fixation *k +1*. We tested if the saccade probabilities among domains were different between good and poor performers by using saccade probability matrices as vectorized features in a linear SVM analysis (other details as described in section on "*SVM classification and feature selection based on gaze metrics*").

Third, we computed the correlation between fixation distributions over images. Each image was divided into 13x18 tiles, and a two-dimensional histogram of fixations was computed for each image and participant. Binning at this resolution yielded non-empty bins for at least 15% of the bins; results reported were robust to finer spatial binning. The vectorized histograms of fixations were correlated between every pair of performers for each image, and median correlations compared across the two categories of performers, with a Wilcoxon signed rank test.

As before, we also compared the median fixation correlations among the good (or poor) performers with the median fixation correlations across good and poor performers (intra- versus inter-category), for all images, with a one-tailed signed rank test.

Fourth, we tested whether good and poor performers fixated on distinct sets of low-level spatial features in the images. For this, we identified spatial features that explained the greatest amount of variance in fixated image patches across good and poor performers. Specifically, image patches of size 112x112 pixels around each fixation point, corresponding to approximately 4° of visual angle were extracted from each image for each participant and converted to grayscale values using the *rgb2gray* function in Matlab, which converts RGB images to grayscale by eliminating the hue and saturation information while retaining the luminance. Two sets of fixated image patches was constructed separately for the good and poor performers. Each of these image patch sets was then subjected to Principal Component Analysis (PCA), using the *pca* function in Matlab, to identify low-level features in the image which occurred at the most common points of fixation across each group of subjects (Fig 3D). We, next, sorted the PCA feature maps based on the proportion of explained variance, and correlated each pair of sorted maps across good and poor performers; in the Results, we report average correlation values across the top 150 principal component maps. We did not attempt an SVM classification analysis based on PCA features, because of the high dimensionality of the extracted PC maps ($\sim 10^4$), and the low number of data points in our experiment ($\sim 800$). We also performed the same analysis after transforming each image into a grayscale saliency map using the frequency tuned salient region detection algorithm [24]. The same analyses were repeated for spatial features extracted from good and poor performers' fixated image patches.

Fifth, we tested if good and poor performers differed in terms of the spatial patterns of their fixations relative to the change region. For this, we computed the fixation frequency (counts) and the total fixation duration for each participant, based on the distance relative to the center of the change location, binned in concentric circular windows of increasing radii, in steps of 50 pixels. Each of these metrics were normalized by the respective parameter for each image and pooled together, separately for the good and poor performers, and compared between the two classes of performers with the Kolmogorov-Smirnov test (S4 Fig). Finally, to test if good and poor performers differed in terms of their latencies to fixate on the change region, we also compared the time to first fixation on the region of change, or the time from trial initiation to detect changes (on successful trials) for good and poor performers (Fig 2G and 2H).

## Model simulations and choice of parameters

The model was simulated with a sequence of operations, as shown in Fig 4B. The model has been fully described in the Results. In these simulations, the CVR transformation that mimics foveal magnification was performed before the saliency map was computed (see next section). This sequence mimics the order of operations observed in the brain: foveal magnification occurs at the level of the retina, whereas saliency computation occurs at the level of higher brain structures like the superior colliculus [66] or the parietal cortex [67]. Saliency maps were computed using the frequency tuned salient region detection algorithm [24]. Because of this sequence of operations, we needed to re-compute the saliency map for each image for every possible location of fixation (at the pixel level): an operation that is computationally unfeasible on a standard desktop system. To expedite the computation, we represented each image in a reduced 864x648 pixel space and divided each image into a grid of non-overlapping patches or regions (72x54; Fig 4B), such that each patch covered 12x12 pixels. For two images of portrait orientation (Images #10 and #19; in Data Availability link), the same operations were done except that x- and y- grid resolutions were interchanged. We then pre-computed CVR

transforms and saliency maps for each of these pre-computed grid centers and performed simulations based on these region-based representations of the images.

Model parameters used for the simulations are specified in Table 1. Model parameters were not fit to human behavioral data, for example, using maximum-likelihood estimation. Rather, we selected model parameters so that they either matched the parameters used in the experiment (e.g. image and blank durations), or matched human metrics. We describe next the specific justification for choice of each model parameter listed in Table 1.

The time bin (Δt) was specified as 25 ms; larger and smaller values resulted in less or more frequent evaluations of the evidence (Eq 1) producing correspondingly faster or slower accumulation of the evidence. The image and blank durations (τ) were fixed at 10 time bins (250 ms), matching their durations in the actual experiment. The trial duration was fixed to 2400 bins (60 s), again matching the actual experiment. The temperature (T) parameter was set to ensure a similar range of saccade amplitude variance in the model, as in the human data. The decay factor (γ), which determines how quickly accumulated evidence "decays" over time, and decay scale (β), which governs the spatial extent of evidence accumulation, were set to default values that enabled the model to match *average* human performance across all images. Then, their values were varied over a wide range to test the effect of these parameters on model success with change detection. The spatial distribution of the decay parameter at each region was specified based on a two-dimensional Gaussian function, with its peak at the region of fixation; therefore, $\gamma_i$ at each location is a function of time and depends on the current region being fixated. Noise scale (W), which controls the noise added during the evidence accumulation process, and threshold ($F_c$), which controls the threshold value of evidence needed for reporting a change (Fig 4D), were set so that their respective values ensured negligibly low false-positive rates (< 2%), overall. The prior odds ratio (P) and "no change" threshold ($F_n$) were set to values that provided an approximate match to the median human fixation durations. Firing rate bounds ($\lambda_{min}$, $\lambda_{max}$) for encoding saliency were between 5 and 120 spikes per time bin. This corresponds to an overall population firing rate range of 0.2–4.8 kHz, which, assuming around 50 units in the neural population encoding each region, works out to a firing rate in the range of 4–96 Hz per neuron; these numbers mimic the biologically-observed range of firing rates for SC neurons (~5–100 Hz, White et al. 2017; their Fig 3). The firing rate prior ($\mu_f$) was set to 3 spikes per bin, and the effect of varying this parameter on performance was also tested (Fig 5C). Finally, we used a third-order Taylor series approximation to the softmax function to achieve a softer saturation of this function. Note that these model parameter values were chosen based on human gaze metrics, or average task performance, but never based on task performance in individual images, to avoid circularity when correlating model performance with human performance across images (see Materials and Methods section "*Comparison of model performance with human data*").

Human saccade sequences tend to be biased in terms of the amplitude of individual saccades, and the angles between successive saccades (S7 Fig); these biases likely reflect properties of the oculomotor system that generates these saccades [33]. Because these saccade properties are not emergent features of our model, we matched the human saccade turn angle and amplitude distributions in the model. This was done by multiplying the map of evidence accumulated with the human saccade amplitude and turn angle distribution, before imposing the softmax function for computing saccade probabilities (Fig 4B). The effect of this bias was that the model generated scan-paths which qualitatively resembled human scan-paths (e.g. Fig 4C). Again, we sought to match only human saccade statistics in the model, and not task performance, when imposing this saccade bias to avoid circularity when computing the correlation between model and human performance in the change blindness task (see Materials and Methods section "*Comparison of model performance with human data*"). We repeated the

simulations without imposing human saccade biases on the model, and obtained nearly identical results.

**Cartesian variable resolution (CVR) transform.** We modeled a key biological feature of visual representations of images, in terms of differences between foveal and parafoveal representations. When a particular region is fixated, the representation of the fixated region, which is mapped onto the fovea, is magnified whereas the representation of the peripheral regions are correspondingly attenuated (S5 Fig). We modeled this using the Cartesian Variable Resolution (CVR) transform, which mimics known properties of visual magnification in humans [26].

The enhanced sensory representation of the foveated (fixated) region was modeled according to the following mathematical transformation of the image. We considered the foveated pixel to be the origin, denoted by $(x_0, y_0)$ in the original image. An arbitrary point in the image, denoted by $(x, y)$ is at a distance from the origin given by, $dx = x-x_0$ and $dy = y-y_0$. The following logarithmic transformation was then performed:

$$dv_x = \ln(\beta dx + 1)S_{fx}; \; dv_y = \ln(\beta dy + 1)S_{fy} \tag{8}$$

where, $\beta$ is a constant (= 0.05) that determines central magnification, and $S_{fx}$ and $S_{fy}$ (= 200) are scaling factors along $x$ and $y$ directions, respectively; results reported were robust to modest variations of these parameter values. The final coordinates of the CVR transformed image are given by: $x_1 = x_0+dv_x$ and $y_1 = y_0+dv_y$.

## Computation of the likelihood ratio ($L_i(t; z)$)

We provide here a detailed derivation of Eq 1 in the Results, involving computation of the likelihood ratio $L_i(t; z)$ for change versus no change at each region $A_i$. At each fixation, the model is faced with evaluating evidence for two hypotheses: change ($C$) versus no change ($N$). Note that the true difference between the firing rates of the generating processes at the change region is not known to the model, *apriori*; this corresponds to the fact that, in our experiment, the observer cannot know the precise magnitude or nature of the change occurring in each change image pair, *apriori*. We posit that the model expects to observe a firing rate difference of $\pm\mu_f$ between the means of the two Poisson processes associated with the change region; this represents the *apriori* expectation of the magnitude of change for human observers. Here, we model this prior as a singleton value, although it is relatively straightforward to extend the model to incorporate priors drawn from a specified density function (e.g. Gaussian).

Let $X^i$ and $Y^i$ denote the number of spikes observed in the m and n−p time-bins that the model fixates on the two images ($A$ or $A'$), respectively (Fig 4A). Let $\lambda_i$ denote the mean firing rate observed during this fixation, up until the current time bin; for this derivation, we posit that $\lambda_i$ is measured in units of spikes per time bin; measuring $\lambda_i$ in units of spikes per second simply requires multiplication by a scalar factor (Table 1), which does not impact the following derivation. The model estimates the mean firing rate over the fixation interval as $\lambda_i = (X^i+Y^i)/(m+n-p)$. Note that this estimate of the mean firing rate is updated during each fixation across time bins.

For hypothesis $C$ to be true, $X^i$ would be a sample from a Poisson process with mean, $\Gamma_1 = m(\lambda_i+\mu_f)$ or $\Gamma_1 = m(\lambda_i-\mu_f)$ and $Y^i$ would be a sample from a Poisson process with mean, $\Gamma_2 = (n-p)(\lambda_i-\mu_f)$ or $\Gamma_2 = (n-p)(\lambda_i+\mu_f)$, respectively. Similarly, for hypothesis $N$ to be true, $X^i$ would be a sample from a Poisson process with mean, $\Gamma_1 = m(\lambda_i)$ and $Y^i$ would be a sample from an identical Poisson process with mean, $\Gamma_2 = (n-p)(\lambda_i)$. For detecting changes, we assume that the model computes only the difference in the number of spikes, $Z^i = Y^i-X^i$, between the two images, rather than keeping track of the precise number of spikes generated by each image. The observed difference $Z^i$ could, therefore, be positive or negative.

For hypothesis C (occurrence of change), the likelihood of observing a specific value of the difference in the number of spikes across the two images $Z^i = z$ is given as:

$$P(Z^i = z | C) = \frac{1}{2} \left( P(Z^i = z | X^i \sim \rho(m(\lambda_i + \mu_f)), Y^i \sim \rho((n-p)(\lambda_i - \mu_f)) \right.$$

$$\left. + P(Z^i = z | X^i \sim \rho(m(\lambda_i - \mu_f)), Y^i \sim \rho((n-p)(\lambda_i + \mu_f))) \right) \quad (9)$$

where $\rho$ denotes the Poisson distribution. Here, we have assumed that the prior probabilities of encountering image *A* or *A'* when the fixation lands in a given region are equal (the 1/2 factor). Similarly, for hypothesis *N* (no change), the likelihood of observing a specific difference in the number of spikes, z, is given as:

$$P(Z^i = z | N) = P(Z^i = z | X \sim \rho(m\lambda_i), Y \sim \rho((n-p)\lambda_i)) \quad (10)$$

The likelihood ratio of hypotheses, change versus no change, is computed as:

$$L_i(z; \; t) = \frac{P(Z^i(t) = z | C)}{P(Z^i(t) = z | N)} \quad (11)$$

We next expand these expressions with the analytical form of the Poisson distribution, $P(X = k; \; X \sim \rho(\lambda)) = \frac{e^{-\lambda}\lambda^k}{k!}$, and marginalize over all values of $X^i = x$ and $Y^i = x+z$. These calculations involve computing an infinite sum which can be efficiently solved using Bessel functions. Specifically, the infinite sum in our calculation can be computed using the identity: $\sum_{y=0}^{\infty} \left( \frac{c^y}{y!(y+z)!} \right) = \sqrt{c}^{-z} I(z.2\sqrt{c})$ where *I* is a modified Bessel function of the first kind.

With some algebra, we can show that:

(i) when $Z_t^i \geq 0$:

$$L_i(T) = \frac{B_1 \sum_{x=0}^{\infty} \left( \frac{(m(n-p)(\lambda_i^2 - \mu_f^2)^x)}{x!(x+z)!} \right)}{\frac{B_2 \sum_{z=0}^{\infty} (m(n-p)\lambda^2)^x}{x!(x+z)!}} = \frac{B_1 c_1^{-z} I_z(2c_1)}{B_2 c_2^{-z} I_z(2c_2)} \quad (12)$$

where $c_1 = \sqrt{m(n-p)(\lambda_i^2 - \mu_f^2)}$ and $c_2 = \sqrt{m(n-p)\lambda_i^2}$,

$$B_1' = 0.5 m^{-z} \begin{bmatrix} (\lambda_i - \mu_f)^{-z} e^{-(m(\lambda_1 + \mu_f) + (n-p)(\lambda_i - \mu_f))} \\ +(\lambda_i + \mu_f)^{-z} e^{-(m(\lambda_i - \mu_f) + (n-p)(\lambda_i + \mu_f))} \end{bmatrix} \text{ and } B_2 = (n-p)^z \lambda_i^z e^{-\lambda_i(m+n-p)}.$$

(ii) when $Z_t^i < 0$:

$$L_i(T) = \frac{B_1' \sum_{x=0}^{\infty} \left( \frac{(m(n-p)(\lambda_i^2 - \mu_f^2)^x)}{x!(x+z)!} \right)}{\frac{B_2' \sum_{z=0}^{\infty} (m(n-p)\lambda^2)^x}{x!(x+z)!}} = \frac{B_1' c_1^z I_z(2c_1)}{B_2' c_2^z I_z(2c_2)} \quad (13)$$

where

$$B_1' = 0.5 m^{-z} \begin{bmatrix} (\lambda_i + \mu_f)^{-z} e^{-(m(\lambda_1 + \mu_f) + (n-p)(\lambda_i - \mu_f))} \\ +(\lambda_i - \mu_f)^{-z} e^{-(m(\lambda_i - \mu_f) + (n-p)(\lambda_i + \mu_f))} \end{bmatrix}, \quad (14)$$

$$B_2 = m^{-z} \lambda_i^{-z} e^{-\lambda_i(m+n-p)}$$

and $c_1$ and $c_2$ are the same as before.

We computed the value of the Bessel function using the Matlab function besseli. When values of x and z were large or disproportionate, Matlab's floating point arithmetic could not compute these expressions correctly; in this case, we employed variable precision arithmetic (*vpa* in Matlab). In addition, d for extreme values of *x* and *z* we adopted the following approximations:

i. For sufficiently large values of x: $I(z; \ x) \approx \frac{e^x}{\sqrt{2\pi x}}\left[1 - \frac{4z^2-1}{8x}\right]$

ii. For sufficiently large values of z: $I(z; \ x) \approx \frac{\left(\frac{x}{2}\right)^z}{\Gamma(z+1)}$

Note that the model makes the following assumptions: (i) the model makes a change versus no-change decision based on the difference of spike counts ($Z_i^t = z$), rather than by keeping track of the absolute spike counts produced by each image (see next); (ii) the model estimates the average firing rate based on the number of spikes produced until that time-bin $\lambda_i = (X_i+Y_i)/(n-p+m)$; (iii) the model has a discrete, single valued, *prior* on the change in firing rates $\mu_\beta$ this prior is different from the actual difference in firing rates across the two images, which is computed based on the difference in their, respective, salience values during the simulation (see also Fig 5C).

## Comparison of model performance with human data

Using the saccade generation model shown in Fig 4B, we simulated the model 100 times using the same images as employed in the human change blindness experiment (Fig 1A). All stochastic parameters (evidence noise, fixation durations) were resampled with fresh random 'seeds' for each iteration of the model. We then computed the accuracy of the model as the proportion of times the model detected the change—fixation on change region until threshold crossing (Fig 4D)—versus the proportion of times the model failed to detect the change region. These proportions of correct detections were then compared for human performance (average across n = 39 participants) versus model performance (n = 100 iterations), across images, using robust correlations [68]. For these analyses, we employed the state-of-the-art DeepGaze II network [17] for generating the saliency map.

Next, we performed control analyses to compare the SPRT model with three other change detection models, each with particular differences in search strategy or stopping rule. First, we tested a model that failed to integrate evidence effectively by setting γ = 1 in the evidence integration step (Eq 2). Such a model completely ignores past evidence and makes decisions based solely on instantaneous posterior odds ratio $(L_i(t)P_i)$. Second, we tested a model with an alternative stopping rule in which the change was detected based on the derivative of the posterior odds ratio (difference of log $(L_i P_i)$ between two successive timesteps) crossing a threshold. For these two models, threshold values for terminating the simulation were determined based on two pilot runs across all 20 images; thresholds were chosen such that the models provided a negligible proportion of false-alarm (<0.01%) comparable with our experimental data. Third, we tested a model in which evidence computation and accumulation were intact, but the model selected the next location of saccade with a random strategy. This was achieved by setting a high value of the temperature parameter (T) in the final softmax function (T = $10^4$), which resulted in a nearly uniform probability, across the image, of selecting the next fixation ("random searcher"). For all three models, we identically matched the timing and distribution of fixation interval durations with our standard SPRT model. The distribution of absolute differences in performance between the human data and our model across images, and the corresponding distributions for control models were compared with paired signed rank tests (Fig 6C).

Finally, we tested the model's ability to predict human gaze shift strategies. For this, we employed the following approach. First, we identified the top 10 fixated clusters in each image. Next, we constructed a saccade probability matrix between every pair of clusters among these ten clusters (Fig 7C, rows/columns 1–10) in the human data, by combining fixation data across all n = 39 participants. The model was then simulated 40 times, and the average probability of saccades between the same clusters for each image was computed for the model. These 10x10 saccade probability matrices were then linearized and compared between the model and human data using Pearson's correlations (Fig 7D, left).

## Comparison of model performance with DeepGaze

We also compared the model's ability to predict human gaze patterns with that of DeepGaze II [17]. DeepGaze is among the top-ranked algorithms for human gaze prediction, and is based on a deep learning model for fixation prediction which employs features extracted from the VGG-19 network, another deep learning neural network trained to identify objects in an image. For this comparison, the model was simulated with all of the same steps as in Fig 4B, except that no likelihood ratio was computed, and no evidence accumulated. Rather, saccades occurred stochastically based on the same softmax rule as employed in our algorithm (Eq 3), but based on DeepGaze II saliency values alone. Again, saccade probability matrices were compared between the DeepGaze II prediction and human data using Pearson's correlations (Fig 7D, right).

To enable a fair comparison with DeepGaze we incorporated the following additional features in the DeepGaze model simulations. First, inhibition-of-return (IOR) is an emergent feature of our model (see Results). We, therefore, incorporated IOR in the DeepGaze model as well [34]. IOR was implemented as a Gaussian patch (G) centered on the current fixation (x, y) with a standard deviation ($\sigma$) of 20 pixels. The amplitude of G was scaled up by a time dependent factor (tanh(0.05 t)), so that the impact of IOR increased progressively over the course of the trial. IOR values were accumulated in a spatial map with a discount factor of 0.25 across successive timesteps (IOR(x, y, t) = (0.25 $^*$ IOR(x, y, t—1)) + G(x, y; $\sigma$)). IOR values were clipped between 0 and 1, and the complement of IOR map was multiplied with the foveally-magnified saliency map before computing the next location of fixation. Second, because the DeepGaze model was not accumulating evidence for change, there was no clear termination criterion. Therefore, we identically matched the timing and distribution of fixation interval durations (timesteps for each fixation) with our SPRT model. This was accomplished by initiating and terminating each fixation in the DeepGaze model at the exact same times when these were initiated or terminated in the SPRT model, respectively. Third, to ensure that both the model and DeepGaze produced saccades with the same level of stochasticity we identically matched the temperature parameter in the softmax function (Eq 3) for deciding the next saccade location. Lastly, we also performed comparisons with the human data by limiting the saccade amplitude range for comparison. The SPRT model (and humans) make many short saccades, whereas DeepGaze primarily makes long saccades (Fig 7B). Therefore, we performed a control analysis, comparing the human data, SPRT model and DeepGaze considering only saccades with amplitudes greater than the 10th percentile of those generated by the DeepGaze model (S8C and S8D Fig).

## Supporting information

**S1 Fig. Re-analysis of gaze metrics by re-classifying good and poor performers based on a median split of performance.** From top to bottom row: Re-analysis of the data shown in Figs 1 and 2 (main text), except that "good" and "poor" performers were defined based on a median

split of the data. Other conventions are the same as in the corresponding figure panels in the main text.
(TIF)

**S2 Fig. Gaze metrics predictive of success, distributions of gaze metrics and variance in success rates across images. A**. Pair-wise correlations among the eight gaze metrics used as features in classification analysis of good versus poor performers (Fig 1C, main text). Gray squares: non-significant correlations. Colored square: significant correlations at p<0.01 with Bonferroni correction for multiple comparisons. Abbreviations are as in Fig 1C (main text). **B.** Saccade amplitude (left) and fixation duration (right) distributions for representative participants (ID-s in each subplot title). Red fits: Mixture of Gaussians model. p-value in title of each subplot indicates significance level for deviation from unimodality per Hartigan's dip test (smaller p-values represent greater evidence of bi/multi-modailty). **C**. Success rates of human observers on the change blindness trial images (n = 20), sorted by the proportion of hits. Error bars denote standard error of the mean performance across participants.
(TIF)

**S3 Fig. Fixated features for good and poor performers. A.** Difference between the average saccade probability matrices for the good and poor performers (good minus poor). Other conventions are the same as in Fig 3A (main text). Note that these differences are 3 orders of magnitude smaller than the values in Fig 3A (main text). **B.** Same as in Fig 3D (main text) except that fixated features were identified following PCA on 112x112 patches extracted from a saliency map, rather than the grayscale image. The saliency map was generated with the frequency tuned saliency algorithm [24]. Other conventions are the same as in Fig 3D main text.
(TIF)

**S4 Fig. Distribution of fixations, relative to change location, for good and poor performers. A.** Distribution of frequency of fixations, binned based on the distance of fixation relative to the center of the change location, separately for good (red) and poor (blue) performers. **B.** Same as in panel A but for the total fixation duration.
(TIF)

**S5 Fig. Mimicking foveation in the model.** Illustration of foveal magnification with the Cartesian Variable Resolution (CVR) transform for a hypothetical fixation (highlighted by the circle) on one of the images used in the change blindness task (Image #6, S1 Table).
(TIF)

**S6 Fig. Dependence of the likelihood ratio (L(t; z)) on mean firing rate and firing rate prior. A.** Likelihood ratio (L(t; z)) as a function of spike count difference between the first and second image (z, Eq 1; main text) for different values of the mean firing rate, $\lambda = 4 \ldots 10$ spikes/bin. The number of time bins for which the first and second images were fixated (m and n−p, respectively) have each been fixed to 5 bins, and the firing rate difference prior, $\mu_f$ fixed at 3 spikes/bin. Curves of progressively lighter shades: increasing values of the mean firing rate. **B.** Same as in A, but for different values of the firing rate difference prior, $\mu_f = 1, 3, 5 \ldots 13$ spikes/bin and mean firing rate $\lambda$ fixed at 40 spikes/bin. Curves of progressively lighter shades: increasing values of $\mu_f$.
(TIF)

**S7 Fig. Mimicking Saccade Turn Angle distribution.** Polar heat map indicating the distribution of human saccade amplitudes and turn angles. The arrow indicates the location of the last saccade. The histogram was computed using data from all (n = 39) participants and all (n = 20) images. The bias against right angled turns is apparent. The distribution was

smoothed both along the radial and angular directions, for display purposes only.
(TIF)

**S8 Fig. Model saccade probability matrices, and correlations with human data (control analyses). A-B.** Same as in Fig 7C and 7D (main text), except with replacing DeepGaze's saliency algorithm with the frequency-tuned salient region detection algorithm. **C-D.** Same as in Fig 7C and 7D (main text) except including only saccades whose amplitude was at least as large (or greater) than the 10th percentile of saccade amplitudes generated by the DeepGaze model (Fig 7B, main text, dashed vertical line). For C, the saccade probability matrix was normalized by its range for visualization purposes only. Other conventions are the same as in Fig 7C and 7D (main text).
(TIF)

**S1 Table. List of images employed in the change blindness task.**
(DOCX)

## Acknowledgments

We thank Ranit Sengupta for help with compiling the results and Guruprasath Gurusamy for help with preparing figures. We also thank Prof. Veni Madhavan for generously sharing the eye tracker employed in these experiments.

## Author Contributions

**Conceptualization:** Devarajan Sridharan.

**Data curation:** Akshay Jagatap, Simran Purokayastha.

**Formal analysis:** Akshay Jagatap, Hritik Jain.

**Funding acquisition:** Devarajan Sridharan.

**Investigation:** Simran Purokayastha.

**Methodology:** Akshay Jagatap, Simran Purokayastha, Hritik Jain.

**Supervision:** Devarajan Sridharan.

**Writing – original draft:** Devarajan Sridharan.

**Writing – review & editing:** Devarajan Sridharan.

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
