## [Decision Letter · Decision Letter 0]

30 Apr 2021

Dear Prof. Sridharan,

Thank you very much for submitting your manuscript "Neurally-constrained modeling of human gaze strategies in a change blindness task­" for consideration at PLOS Computational Biology.

As with all papers reviewed by the journal, your manuscript was reviewed by members of the editorial board and by several independent reviewers. In light of the reviews (below this email), we would like to invite the resubmission of a significantly-revised version that takes into account the reviewers' comments.

We cannot make any decision about publication until we have seen the revised manuscript and your response to the reviewers' comments. Your revised manuscript is also likely to be sent to reviewers for further evaluation.

Sincerely,

Alireza Soltani

Associate Editor

PLOS Computational Biology

Wolfgang Einhäuser

Deputy Editor

PLOS Computational Biology

Reviewer's Responses to Questions

**Comments to the Authors:**

**Please note that the review by reviewer #2 is uploaded as an attachment.**

Reviewer #1: Authors have done an interesting study on the pattern of saccades for detection of change in consecutive images and showed that there is a variability in the performance of subjects. besides they provided a model based on evidence accumulation that they claimed has better performance than Deep gaze neural networks.

I believe that though the authors did a very good job in their analyses and modeling, but there is some gaps between behavioral model and study that should be addressed. my suggested extra analyses can be listed as below:

1- in behavioral analysis, though they clustered saccade related areas, but at each image, different clusters are associated with change location, so, we can't infer how much subjects fixated on areas near to change location. so please add an extra analysis that show good performers and poor performers how much fixated in areas near to change location; it can be shown by the total fixation duration and frequency at different locations based on the distance of fixation location relative to the center of change location.

2- the time to first saccade to change location and also the time to fixate 3 sec on change area can be provided for each image and each subject. it can help to analyze the behavior relative to model; because if we see poor performers fixate on change location but don't detect the change, it means that fixation duration or drift rate were not enough for change detection or their threshold was high; but if they did not fixate on areas near change location it means that their weakness is on finding relevant area.

3- one hypothesis can be that the threshold is decaying with different rate in different subjects and so subjects with lower decaying fixate more on each location, so it will be great if you can add threshold decaying that can account for an important mechanism in the model for poor performers.

4- in feature list, saccade amplitude variance should be divided to its components; in other words, because saccade amplitude distribution probably is bimodal and short and long saccades have different means and also different probability, using variance cause to miss important information on the strategy of subjects, so please fit distribution of saccade amplitude by gaussian mixture model and investigate the mean of each cluster and their frequency. please check bimodality for fixation duration too, if it is bimodal the same issue should be chacked for fixations before large saccade and small saccades.

5- study has two parts, first part studied individual differences that is very important and can extend this study for many applications such as clinical applications, but in modeling part analyses are mostly n images' difficulty (performance on each image). so the coherence of paper has been reduced by missing the link between model and individual differences; I understand that data is small and fitting on each subject has some difficulties, but using hierarchical bayesian modeling you can use data from all subjects and have two clusters of poor and good performers on that, and provide the value of each parameter for each subject, it will help to analyze what aspects of model caused to good or poor performance in different subjects. this will help to make paper more coherent and will help others to find important applications for your model to study individual differences.

6- to be more fair on deep gaze, it is good to compare it based on just large saccades, because it seems that deep gaze is good in predicting large saccades and not small saccades.

minor points:

in some of images your change is related to color change, but you have just mentioned normal vision, please add that they have not any color blindness too.

in fig. S1, you said that circle highlighted... while in figure you have square for that.

in the model, inhibition of return has not been included, it seems that it may help to enhance the model; as a suggestion if you'd like you can test it too.

Reviewer #2: See attachment

**Have the authors made all data and (if applicable) computational code underlying the findings in their manuscript fully available?**

Reviewer #1: Yes

Reviewer #2: Yes

PLOS authors have the option to publish the peer review history of their article (what does this mean?). If published, this will include your full peer review and any attached files.

Reviewer #1: **Yes: **Abdol-Hossein Vahabie

Reviewer #2: No
---

## [Decision Letter · Decision Letter 1]

4 Aug 2021

Dear Prof. Sridharan,

We are pleased to inform you that your manuscript 'Neurally-constrained modeling of human gaze strategies in a change blindness task­' has been provisionally accepted for publication in PLOS Computational Biology.

**Also, Reviewer # 2 has a final useful suggestion that could be included in the final manuscript.**

Best regards,

Alireza Soltani

Associate Editor

PLOS Computational Biology

Wolfgang Einhäuser

Deputy Editor

PLOS Computational Biology

Reviewer's Responses to Questions

**Comments to the Authors:**

Reviewer #1: I am satisfied with the revision.

Reviewer #2: The authors did a great job addressing my concerns. I especially like the higher-level engagement with mechanisms both in the results and the discussion. I very much look forward to seeing the article in print.

One comment: In figure 1D, several measures are listed for feature importance. It seems a bit strange to label one "Treebagger," as it is a specific implementation of an algorithm. For example, Fisher score, info gain and AUC change are all mathematical technics, while Treebagger is a proprietary implementation of a machine learning method. It might be better to label it as the exact method Treebagger utilized to compute feature importance (usually change in OOB Error, as described in the methods), rather than the algorithm.

**Have the authors made all data and (if applicable) computational code underlying the findings in their manuscript fully available?**

Reviewer #1: None

Reviewer #2: Yes

PLOS authors have the option to publish the peer review history of their article (what does this mean?). If published, this will include your full peer review and any attached files.

Reviewer #1: **Yes: **Abdol-Hossein Vahabie

Reviewer #2: No

---

## [Editor Report · Acceptance letter]

19 Aug 2021

PCOMPBIOL-D-21-00403R1 

Neurally-constrained modeling of human gaze strategies in a change blindness task­

Dear Dr Sridharan,

I am pleased to inform you that your manuscript has been formally accepted for publication in PLOS Computational Biology. Your manuscript is now with our production department and you will be notified of the publication date in due course.

With kind regards,

Zsofi Zombor
